# A Semantic Invariant Robust Watermark for Large Language Models

**Aiwei Liu[1], Leyi Pan[1], Xuming Hu[2], Shiao Meng[1], Lijie Wen[1] ***,
[1]School of Software, BNRist, Tsinghua University
[2]The Hong Kong University of Science and Technology (Guangzhou)
`liuaw20@mails.tsinghua.edu.cn, wenlj@tsinghua.edu.cn,`

## Abstract

Watermark algorithms for large language models (LLMs) have achieved extremely high accuracy in detecting text generated by LLMs. Such algorithms typically involve adding extra watermark logits to the LLM's logits at each generation step. However, prior algorithms face a trade-off between attack robustness and security robustness. This is because the watermark logits for a token are determined by a certain number of preceding tokens; a small number leads to low security robustness, while a large number results in insufficient attack robustness. In this work, we propose a semantic invariant watermarking method for LLMs that provides both attack robustness and security robustness. The watermark logits in our work are determined by the semantics of all preceding tokens. Specifically, we utilize another embedding LLM to generate semantic embeddings for all preceding tokens, and then these semantic embeddings are transformed into the watermark logits through our trained watermark model. Subsequent analyses and experiments demonstrated the attack robustness of our method in semantically invariant settings: synonym substitution and text paraphrasing settings. Finally, we also show that our watermark possesses adequate security robustness. Our code and data are available at https://github.com/THU-BPM/Robust_Watermark.

## 1 Introduction

As the quality of text generated by large language models (LLMs) continues to improve, it addresses a multitude of practical challenges on one hand, while simultaneously giving rise to a spectrum of new issues on the other. Specifically, the proliferation of LLM-generated text on the Internet may lead to an influx of rumor-based content and text copyright concerns (Rillig et al., 2023). Therefore, the detection and labeling of machine-generated text have become extremely important.

Text watermarking techniques for LLMs usually embed specific information during text generation to allow high-accuracy detection of LLM-generated text. The mainstream approach for embedding such information is to add extra watermark logits on top of the logits generated by the LLM. For example, Kirchenbauer et al. (2023a) divide the vocabulary into red and green lists and increase the scores for the green tokens as the watermark logits. However, current watermarking algorithms cannot possess both attack robustness (robustness to modifications of the watermarked text) and security robustness, which refers to the difficulty of inferring watermarking rules from watermarked text. For example, Zhao et al. (2023) demonstrates that global fixed watermark logits enhance attack robustness, yet they compromise security due to vulnerability in word frequency analysis (Sadasivan et al., 2023). This is because the frequency of tokens from their green list is much higher compared to those in normal text, the high-frequency tokens could be simply treated as green tokens and further used to remove the watermark. Essentially, in current watermark algorithms, the watermark logits for each token depend on its preceding tokens. As the number of required preceding tokens increases, watermarking complexity rises, leading to reduced attack robustness but increased security robustness.

To resolve the aforementioned trade-off, we propose a semantic invariant watermarking algorithm that achieves reasonable attack and security robustness. The core motivation is generating watermark

---
*Corresponding author

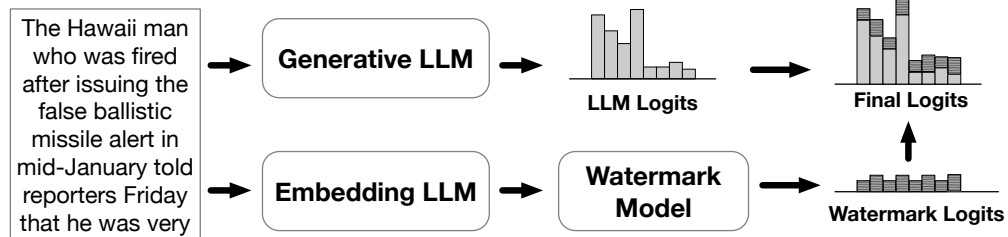

Figure 1: An illustration of our semantic invariant robust watermarking method. Text is input into a generative LLM for token logits and an embedding LLM for text embedding. The embedding is converted into watermark logits via the Watermark Model. LLM logits and watermark logits are then combined for final logits, which decode the next token using any method.

logits for each token based on the preceding tokens' semantics rather than their token IDs. Thus, semantically invariant text modifications do not alter the watermark logits, while the diversity of text semantics increases watermark complexity and guarantees security against watermark cracking. Specifically, to extract semantically invariant features, we use an auxiliary LLM encoder (e.g., BERT) to extract semantic embeddings of the preceding tokens. We then train a small watermark network to transform these embeddings into corresponding watermark logits. The training objective of the watermark network is to ensure a high correlation between the similarities of the output watermark logits and input text embeddings. Also, it's imperative that the watermark logits exhibit sufficient diversity and are unbiased for each token. To achieve these goals, two training objectives are adopted: a similarity loss and a normalization loss. For the similarity loss, to ensure the diversity of the watermark logits, we first rescale the similarity values between text embeddings to range from -1 to 1, and then make the similarity of the generated watermark logits fit this similarity. To ensure unbiased token selection and achieve bimodal scores of the watermark logits, our normalization loss centers the mean of each row and column within a batch of watermark logits to zero, while making the absolute value of each entry as close as possible. During the detection phase, we compute the watermark logits for each token at its position and obtain the corresponding value. We then average the values across all tokens and determine the presence of a watermark by checking if the average is significantly greater than zero.

In the experiment, we evaluate the attack robustness of our watermarking algorithm against various semantically invariant perturbations, including text paraphrasing and synonym replacement. Overall, our watermark robustness is comparable to KGW-1 (global watermark logits), which is close to the robustness upper bound achievable by watermark logit-based methods. Additionally, employing the spoofing attack paradigm used in Sadasivan et al. (2023), we evaluate the decryption accuracy of various watermarking methodologies to gauge security robustness. Our algorithm demonstrates favorable security robustness metrics, effectively resolving the previously encountered trade-off between attack and security robustness. Importantly, the watermark logits could be generated in parallel with the LLM logits, resulting in only a marginal latency increase during text generation.

Our contributions are as follows: **(1)** We propose the first semantically invariant robust watermarking algorithm, which effectively detects watermarks under various semantically invariant perturbations. **(2)** Our algorithm successfully navigates the trade-off between attack robustness and security robustness that plagued previous methods, achieving high performance in both dimensions. **(3)** We propose a watermark model that adeptly transforms semantic embeddings into watermark logits.

## 2 RELATED WORK

Currently, there are two approaches to watermarking text Liu et al. (2023c): post-processing after text generation, and incorporating watermarks during the text generation of LLM.

Post-processing methods make minor semantic-preserving alterations to the text, most commonly lexical substitution-based approaches. For instance, Qiang et al. (2023) embedded watermark information by employing a paraphraser-based lexical substitution, ensuring the text's original semantics remained intact. Yoo et al. (2023) began by selecting stable words to position the watermark and then

used a masked language model to generate candidate words, thereby facilitating the embedding and extraction of watermarks. Munyer & Zhong (2023) generated candidate texts through word2vec-based synonym replacements, subsequently comparing semantic similarities among candidates to finalize the watermarked text. Nonetheless, replacing individual words while only considering the unchanged meaning of those words risks degrading overall text quality. In contrast, generative modification methods like Abdelnabi & Fritz (2021) directly generate watermarked text, along with corresponding networks to recover the watermark information.

Based on the stage at which watermark information is embedded, current watermarking methods for LLMs can be divided into two categories: adding watermarks during token sampling and during logits computation. Christ et al. (2023) proposed a method that embeds watermark information by presetting a random number sequence for token sampling, and detects watermarks by computing correlation between texts and the random number sequence. To improve robustness against text modifications, Kuditipudi et al. (2023) adopted Levenshtein distance for matching texts and the random number sequence, allowing watermarked texts to be detectable even after certain modifications. However, although such token sampling-based watermarking methods minimize the impact on text quality, they also greatly limit the randomness of generated texts, and only work for sampling decoders instead of other decoding methods like beam search. In contrast, another category is to add watermarks during logits computation. Kirchenbauer et al. (2023a) made minor modifications to current token logits based on hashes of previous k tokens (watermark logits). Lee et al. (2023) designed a similar watermarking method for low-entropy code generation. Wang et al. (2023) enabled multi-bit watermark information via a proxy LLM. Liu et al. (2023b) implemented watermarking through parameter sharing between the watermark generator and detector. In this work, we are more concerned with watermark robustness. Zhao et al. (2023) proved that using global fixed watermark logits (k=0) in Kirchenbauer et al. (2023a)'s method achieves very strong attack robustness, yet Sadasivan et al. (2023)'s work shows that lower k values make watermarks vulnerable to security robustness (easy to break). In this work, we propose a semantically invariant robust watermarking scheme, where watermark logits are generated based on semantic information of the text.

## 3 PRELIMINARIES

We first introduce the necessary concepts used in this work. Language models could be divided into generative language models and embedding language models. A generative language model M takes a prompt $x^{prompt}$ and already generated text $\boldsymbol{t}_{:l-1}$ as input and generates the logits for the next token: $P_M(x^{prompt}, \boldsymbol{t}_{:l-1})$. Meanwhile, an embedding language model E could generate an embedding $\mathrm{E}(\boldsymbol{t})$ for the text $\boldsymbol{t}$. Usually, semantically similar texts will generate similar embeddings. For convenience in later sections, we use the language model to refer to generative language models.

A watermarking algorithm for large models embeds specific information in the generated text. The paradigm adopted in this paper is adding small watermark logits to the already generated next token logits. Specifically, the watermark logits can be defined as $P_{\mathrm{W}}(\boldsymbol{x}^{prompt}, \boldsymbol{t}_{:l-1})$, and the final logits could be defined as $P_{\hat{\mathrm{M}}}(\boldsymbol{x}^{prompt}, \boldsymbol{t}_{:l-1}) = P_{\mathrm{M}}(\boldsymbol{x}^{prompt}, \boldsymbol{t}_{:l-1}) + P_{\mathrm{W}}(\boldsymbol{x}^{prompt}, \boldsymbol{t}_{:l-1})$, where $\hat{\mathrm{M}}$ is the watermarked LLM. The watermark detector $P_{\mathrm{D}}$ corresponds to $P_{\mathrm{W}}$, outputting 1 if text $\boldsymbol{t}$ contains the watermark, otherwise 0. for consistency, we explicitly define $P_{\mathrm{W}}^{(i)}$ to denote the value at the i-th position of the watermark logits, and use $P_{\mathrm{W}_i}$ to denote a specific example of watermark logits.

The robustness of our work encompasses two aspects: attack robustness and security robustness. Attack robustness evaluates the probability that the watermarked text can still be correctly identified after semantically invariant modifications. Security robustness evaluates the accuracy of inferring the watermark rules from the watermarked text. Once the watermarking rules are disclosed, users can easily modify the text to remove the watermark, rendering attack robustness meaningless. In our work, security robustness is evaluated by employing spoofing attacks Sadasivan et al. (2023), specifically by conducting statistical analysis on watermarked texts.

## 4 PROPOSED METHOD

In this section, we provide a detailed introduction to the proposed semantically invariant robust watermark algorithm named SIR. First, in Section 4.1, we introduce the overall process of watermark

generation. Then, in Section 4.2, we explain the training process of the watermark model. Finally, in Section 4.3, we present the watermark detection process.

## 4.1 WATERMARK GENERATION

As discussed in the previous sections, one of the most important steps in the watermark generation process is the generation of the watermark logits. The watermark logits for the current token are usually determined by its preceding tokens. Our goal is to construct a continuous and robust mapping from the semantics of the preceding tokens to the watermark logits. In this way, semantic-preserving modifications to the text will only cause small perturbations to the watermark logits.

To extract semantic-invariant features of the text, we utilize an embedding language model E (e.g. BERT). Specifically, given a sequence of preceding tokens $\boldsymbol{t}_{:l-1}$, we first obtain its semantic embedding $\boldsymbol{e}_l = \mathrm{E}(\boldsymbol{t}_{:l-1})$. To transform this semantic-invariant feature to a score over the vocabulary V, we train a specialized watermark model T to generate the watermark logits: $P_{\mathrm{W}} = \mathrm{T}(\boldsymbol{e}_l)$. The overall algorithm is described in detail in Algorithm 1. A thorough introduction to the watermark model will be provided in Section 4.2.

---

**Algorithm 1** Watermark Generation

---

1: **Input:** watermark strength $\delta$, a language model M, previous generated text $\boldsymbol{t} = [t_0....t_{l-1}]$, a text embedding language model E, a trained watermark model T.
2: Generate the next token logits from $P_{\mathrm{M}}$: $P_{\mathrm{M}}(\boldsymbol{x}^{prompt}, \boldsymbol{t}_{:l-1})$.
3: Generate sentence embedding $\boldsymbol{e}_l = \mathrm{E}(\boldsymbol{t}_{:l-1})$.
4: Generate watermark logit $P_{\mathrm{W}}$ from trained watermark model $\mathrm{T}(\boldsymbol{e}_l)$.
5: Define a new language model $\hat{\mathrm{M}}$ where given input $\boldsymbol{t} = [t_0....t_{l-1}]$, the resulting logits satisfy

$$P_{\hat{\mathrm{M}}}(\boldsymbol{x}^{prompt}, \boldsymbol{t}_{:l-1}) = P_{\mathrm{M}}(\boldsymbol{x}^{prompt}, \boldsymbol{t}_{:l-1}) + \delta \times P_{\mathrm{W}}(\boldsymbol{x}^{prompt}, \boldsymbol{t}_{:l-1}).$$

6: **Output:** watermarked next token logits $P_{\hat{\mathrm{M}}}(t_l)$.

---

## 4.2 WATERMARK MODEL

The goal of the watermark model is to convert semantic embeddings into watermark logits. First, we describe several desired properties for the similarity of watermark logits: **semantic-consistent broad range**, **unbiased token preference**, and **balanced score**.

To ensure the attack robustness, the similarity of the watermark logits should have a **semantic-consistent broad range**. That is, the similarity between generated watermark logits should be highly correlated with the similarity between text embeddings. Also, to ensure diversity and security, the similarty should have a broad range between 1 and -1 (language models typically generate embeddings with similarities concentrated between 0 and 1), as shown in the following equation:

$$\forall x, y \in [-1, 1], x < y, \exists i, j : \frac{P_{\mathrm{W}_i} \cdot P_{\mathrm{W}_j}}{||P_{\mathrm{W}_i}||_2 \times ||P_{\mathrm{W}_j}||_2} \in [x, y]. \tag{1}$$

Additionally, to further ensure the security robustness of the watermark, the watermark logits should have an **unbiased token preference**, meaning there should be no statistical preference for any token:

$$\forall i \in \{1, 2, \ldots, |\mathrm{V}|\}, \sum_j P_{\mathrm{W}_j}^{(i)} = 0. \tag{2}$$

Moreover, to make the watermark stable and easy to detect, the watermark logits should have a **balanced score**, meaning the mean of the watermark logits is 0 and the score is uniform relative to the mean value, which is shown in the following equation:

$$\forall j, \sum_{i=0}^{|\mathrm{V}|} \mathrm{sign}(P_{\mathrm{W}_j}^{(i)}) = 0, \tag{3}$$

where $\mathrm{sign}(x) = 1$ if $x$ is greater than 0, and -1 if $x$ is less than 0.

To achieve the above design goals, we devised a non-linear neural network (watermark model) to generate the watermark logits. This neural network consists of several fully connected layers with Rectified Linear Unit (ReLU) activations introduced to induce non-linearity. The watermark model employs two losses to meet the stated objectives: a similarity loss and a normalization loss.

The objective of the similarity loss is to attain a semantically consistent broad-range similarity for the watermark logits. Specifically, in order to achieve a broad range, the similarity of the text embeddings first needs to be normalized to $[-1, 1]$ by calculating the mean of the similarity and then expanding the range using the $\tanh$ function. Finally, the overall goal of the similarity loss is to fit the similarity between the watermark logits to the transformed text embedding similarities. The standard definition of the similarity loss $\mathcal{L}_s$ is as follows:

$$\sum_i \sum_j |\frac{T(\boldsymbol{e}_i) \cdot T(\boldsymbol{e}_j)}{||T(\boldsymbol{e}_i)||_2 \times ||T(\boldsymbol{e}_j)||_2} - \tanh(k_1(\frac{\boldsymbol{e}_i \cdot \boldsymbol{e}_j}{||\boldsymbol{e}_i||_2 \times ||\boldsymbol{e}_j||_2} - \sum_k \sum_l \frac{\boldsymbol{e}_k \cdot \boldsymbol{e}_l}{|N|^2 ||\boldsymbol{e}_k||_2 \times ||\boldsymbol{e}_l||_2}))|, \quad (4)$$

where $|N|$ is the size of the sentence embedding space, T is the watermark model, and $k_1$ is a hyperparameter used to adjust the range of similarity.

We then introduce the normalization loss, whose goal is to have the watermark logits exhibit an unbiased token preference while maintaining a balanced score. Specifically, this requires the mean of each watermark logit to be zero, and the mean over all tokens in the watermark logits to also be zero. Finally, to make the score of the watermark logits uniform, we constrain the absolute value of each value to be close, with the standard definition of the normalization loss as follows:

$$\mathcal{L}_n = \sum_i |\sum_j T(\boldsymbol{e}_i)^{(j)}| + \sum_i |\sum_j T(\boldsymbol{e}_j)^{(i)}| + \lambda_1 \sum_i \sum_j |R - T(\boldsymbol{e}_j)^{(i)}|, \quad (5)$$

Where R is a hyperparameter denoting the target absolute value for each value in the watermark logits, and $\lambda_1$ is a weight conditioning the normalization loss internally.

The final training loss is a weighted sum of the similarity loss and normalization loss, expressed as follows, where $\lambda_2$ is a hyperparameter balancing the two losses:

$$\mathcal{L} = \mathcal{L}_s + \lambda_2 \mathcal{L}_n. \quad (6)$$

To augment the separability of watermark logits, the output from the watermark model is further processed by the $\tanh$ function, yielding the final watermark logits, denoted as: $\tanh(k_2 T(\mathbf{e}_i))$. After this procedure, the values of watermark logits are almost exclusively 1 or -1. This correlates with the concept of red-green tokens in the work of Kirchenbauer et al. (2023a) (denoted as KGW), indicating a fundamental similarity in principles between our approach and the KGW algorithms.

## 4.3 WATERMARK DETECTION

Similar to some previous works Kirchenbauer et al. (2023a), we assume the null hypothesis and compute a z-statistic. We reject the null hypothesis and detect the watermark if the average watermark logit for each token is greater than zero. The watermark logit for each token is calculated via Algorithm 1 and its average is zero and standard variation is one (as shown in Figure 2(c)), which could be represented as:

$$z = \frac{\sum_{j=1}^N (P_W^{(t_j)}(x^{prompt}, \boldsymbol{t}_{:j-1}) - 0)}{N * 1} = \frac{\sum_{j=1}^N P_W^{(t_j)}(x^{prompt}, \boldsymbol{t}_{:j-1})}{N}. \quad (7)$$

Without a watermark, the expected score is 0 since the watermark logit mean is 0. When a watermark is present, the score substantially exceeds 0, making this a suitable validation approach. In our work, the values of watermark logits are almost exclusively 1 or -1, which corresponds to the concept of green-red tokens by Kirchenbauer et al. (2023a). Consequently, our detection methodology is fundamentally the same to theirs, a point we elaborate on in greater detail in the appendix D. Note that the watermarking model is optimized so that the null hypothesis is approximately true on non-watermarked text, but no guarantee can be provided (e.g. on out-of-domain text).

## 5 ROBUSTNESS ANALYSIS

This section analyzes the attack robustness of the watermark algorithm. As detection relies on the z-score, robustness can be assessed by the z-score change with modifications to the generated text $\boldsymbol{t}_{:N}$. The z-score change magnitude indicates robustness. We use $\boldsymbol{t}'_{:N}$ to denote the modified text.

Let $U$ represent the set of altered tokens Their new scores are considered zero. For unmodified tokens, score changes result from embedding alterations of the proceeding tokens:

$$|\Delta z| \leq \frac{\sum_{j \in U} |P_{\mathrm{W}}(\boldsymbol{x}^{prompt}, \boldsymbol{t}_{:j-1})| + \sum_{j \notin U} |P_{\mathrm{W}}(\boldsymbol{x}^{prompt}, \boldsymbol{t}_{:j-1}) - P_{\mathrm{W}}(\boldsymbol{x}^{prompt}, \boldsymbol{t}'_{:j-1})|}{N}. \quad (8)$$

Since watermark logit generation from text embeddings is continuous, the Lipschitz constant L can bound the inequality per properties of continuous functions:

$$|\Delta z| \leq \frac{\sum_{j \in U} |P_{\mathrm{W}}(\boldsymbol{x}^{prompt}, \boldsymbol{t}_{:j-1})| + \sum_{j \notin U} L|\mathrm{E}(\boldsymbol{t}_{:j-1}) - \mathrm{E}(\boldsymbol{t}'_{:j-1})|}{N}. \quad (9)$$

This shows the watermark is theoretically robust as long as text modifications do not drastically alter semantics, such that embeddings stay similar. In fact: significant semantic changes would make the watermark meaningless. Moreover, an analysis of security robustness is provided in the appendix.

## 6 EXPERIMENT

### 6.1 EXPERIMENT SETTINGS

**Dataset and Prompt**: Similar to the previous works Kirchenbauer et al. (2023a), we utilize the C4 dataset (Raffel et al., 2020) for data generation, taking the first 30 tokens as prompts and generating the next 200 tokens. The original C4 texts serve as human-written examples. The test objective is to distinguish between generated text and human-written text. During training of the watermark model, we utilize the WikiText-103 dataset (Merity et al., 2016) (different from C4) to generate embeddings.

**Baseline and Language Model**: We selected two watermarking algorithms as baselines. One is **KGW-k** (Kirchenbauer et al., 2023a) where k is the number of preceding tokens to hash. The other is exponential minimum sampling (**EXP-edit**) (Kuditipudi et al., 2023), an approach that introduces watermarks through a pre-selected sequence of sampling probabilities during the sampling process and adopts edit distance-based detection to achieve strong robustness. For language models, we use LLaMA-7B (Touvron et al., 2023), OPT1.3B, and OPT2.7B (Zhang et al., 2022) for text generation. Additionally, we tested the efficacy of the watermarking algorithm under both stochastic and deterministic scenarios using sampling and beam search decoding algorithms, respectively. For embedding language models, we utilized Compositional-BERT (Chanchani & Huang, 2023) due to its superior ability to produce embeddings that better distinguish text.

**Evaluation**: Similar to Zhao et al. (2023)'s method, to avoid the impact of detection thresholds, we set false positive rates at 1% and 10% and adjusted the detector's thresholds accordingly. For comparison, we also report F1 scores at optimal thresholds. Further, we use the superior LLaMA-13B model for perplexity evaluation. For safety robustness assessment, we adopt Sadasivan et al. (2023)'s approach of analyzing occurrence frequencies of 181 common words. See Section 6.3 for details.

**Hyper-parameters**: The watermark model uses a four-layer fully connected residual network with rectified linear unit activations. Hyperparameters are set to $k_1 = 20$, $k_2 = 1000$, $\lambda_1 = 10$, $\lambda_2 = 0.1$, and the Adam optimizer (lr=1e-5) is used for training. The detailed network architecture is provided in the appendix. All experiments were conducted using the NVIDIA Tesla V100 32G GPU.

### 6.2 ATTACK ROBUSTNESS ANALYSIS

In Table 1, we list the detection accuracy of our watermark algorithm and various baseline algorithms under the no-attack setting and when texts are rewritten using GPT3.5, two different DIPPER settings (Krishna et al., 2023) and the copy-paste attack Kirchenbauer et al. (2023b). For GPT3.5, we use the *gpt-3.5-turbo-0613* version with the prompt *Rewrite the following paragraph:*. For DIPPER-1 the lex diversity is 60 without order diversity, and for DIPPER-2 we additionally increase the order diversity by 20. For copy-paste attack, we insert 600 tokens from the origin text before the generated text.

Table 1 shows that our watermarking algorithm achieves strong robustness against all attacks. Specifically, for watermarked texts rewritten by GPT-3.5 and two DIPPER rewriters, our algorithm still

Table 1: We compared the performance of our watermarking method with others, including KGW-k (Kirchenbauer et al., 2023a) and EXP (Kuditipudi et al., 2023), using text generated by LLaMA-7B. Tests involved watermark detection accuracy under no attack, GPT3.5 rewrite attacks, two DIPPER (Krishna et al., 2023) settings, the copy-paste attack as well as the emoj attack (Appendix G).

| Setting | Method | Sampling | | | | | Beam search | | | | |
| | | 1% FPR | | 10% FPR | | Best | 1% FPR | | 10% FPR | | Best |
| | | TPR | F1 | TPR | F1 | F1 | TPR | F1 | TPR | F1 | F1 |
|---|---|---|---|---|---|---|---|---|---|---|---|
| No attack | KGW-1 | 0.960 | 0.975 | 1.000 | 0.952 | 0.983 | 1.000 | 0.995 | 1.000 | 0.952 | 0.998 |
| | KGW-2 | 1.000 | 0.995 | 1.000 | 0.952 | 0.998 | 1.000 | 0.995 | 1.000 | 0.952 | 1.000 |
| | KGW-4 | 1.000 | 0.995 | 1.000 | 0.952 | 0.998 | 1.000 | 0.995 | 1.000 | 0.952 | 1.000 |
| | EXP-Edit | 0.995 | 0.993 | 1.000 | 0.952 | 0.995 | × | × | × | × | × |
| | SIR(ours) | 1.000 | 0.995 | 1.000 | 0.952 | 0.995 | 1.000 | 0.995 | 1.000 | 0.952 | 1.000 |
| GPT3.5 | KGW-1 | 0.590 | 0.738 | 0.885 | 0.891 | 0.905 | 0.890 | 0.937 | 0.965 | 0.935 | 0.955 |
| | KGW-2 | 0.535 | 0.693 | 0.760 | 0.817 | 0.823 | 0.655 | 0.787 | 0.795 | 0.839 | 0.865 |
| | KGW-4 | 0.225 | 0.364 | 0.490 | 0.614 | 0.705 | 0.420 | 0.587 | 0.660 | 0.750 | 0.795 |
| | EXP-Edit | 0.435 | 0.602 | 0.645 | 0.739 | 0.775 | × | × | × | × | × |
| | SIR(ours) | 0.740 | 0.856 | 0.865 | 0.880 | 0.900 | 0.805 | 0.887 | 0.945 | 0.924 | 0.938 |
| DIPPER-1 | KGW-1 | 0.715 | 0.829 | 0.940 | 0.922 | 0.930 | 0.930 | 0.959 | 0.975 | 0.939 | 0.962 |
| | KGW-2 | 0.450 | 0.616 | 0.710 | 0.785 | 0.815 | 0.770 | 0.865 | 0.880 | 0.888 | 0.908 |
| | KGW-4 | 0.220 | 0.358 | 0.545 | 0.627 | 0.728 | 0.380 | 0.547 | 0.765 | 0.820 | 0.843 |
| | EXP-Edit | 0.630 | 0.768 | 0.740 | 0.804 | 0.830 | × | × | × | × | × |
| | SIR(ours) | 0.765 | 0.862 | 0.905 | 0.903 | 0.920 | 0.890 | 0.937 | 0.950 | 0.927 | 0.948 |
| DIPPER-2 | KGW-1 | 0.765 | 0.862 | 0.935 | 0.918 | 0.925 | 0.910 | 0.948 | 0.975 | 0.940 | 0.960 |
| | KGW-2 | 0.470 | 0.635 | 0.685 | 0.768 | 0.803 | 0.725 | 0.838 | 0.860 | 0.878 | 0.898 |
| | KGW-4 | 0.150 | 0.259 | 0.475 | 0.603 | 0.718 | 0.315 | 0.475 | 0.645 | 0.739 | 0.783 |
| | EXP-Edit | 0.485 | 0.649 | 0.635 | 0.732 | 0.775 | × | × | × | × | × |
| | SIR(ours) | 0.875 | 0.928 | 0.92 | 0.911 | 0.931 | 0.870 | 0.926 | 0.940 | 0.922 | 0.940 |
| Copy-Paste | KGW-1 | 0.854 | 0.901 | 0.897 | 0.918 | 0.920 | 0.923 | 0.934 | 0.918 | 0.929 | 0.943 |
| | KGW-2 | 0.860 | 0.907 | 0.898 | 0.905 | 0.912 | 0.905 | 0.913 | 0.932 | 0.928 | 0.940 |
| | KGW-4 | 0.877 | 0.899 | 0.910 | 0.910 | 0.911 | 0.897 | 0.931 | 0.934 | 0.932 | 0.936 |
| | SIR(ours) | 0.856 | 0.901 | 0.870 | 0.905 | 0.918 | 0.883 | 0.913 | 0.931 | 0.927 | 0.938 |
| Emoj | KGW-1 | 0.973 | 0.983 | 1.000 | 0.952 | 0.990 | 1.000 | 0.995 | 1.000 | 0.952 | 0.995 |
| | KGW-2 | 0.006 | 0.434 | 0.345 | 0.479 | 0.532 | 0.005 | 0.398 | 0.298 | 0.478 | 0.529 |
| | KGW-4 | 0.004 | 0.456 | 0.389 | 0.467 | 0.497 | 0.003 | 0.401 | 0.314 | 0.453 | 0.504 |
| | SIR(ours) | 0.969 | 0.981 | 1.000 | 0.952 | 0.986 | 0.982 | 0.989 | 1.000 | 0.952 | 0.991 |

achieves an average detection F1 score of 0.93. This performance is comparable to that of the KGW-1 algorithm, markedly surpassing other watermark algorithms, including KGW-2, KGW-4, and EXP-Edit. As modifications to the token in the KGW-1 only affect the label of that particular token, this represents the upper bound of attack robustness for watermark algorithms based on watermark logits. Nonetheless, subsequent experiments demonstrate that the KGW-1 algorithm possesses poor security robustness. We also achieve robustness against copy-paste attacks similar to that of the KGW series methods. Appendix G details the robustness to copy-paste and other types of attacks.

To further demonstrate the robustness of our watermarking method against semantic-preserving text attacks, Figure 2(a) compares the detection F1 score of our watermark algorithm and other baseline methods under different synonymous word substitution scenarios. The synonyms are obtained from the WordNet synset (Miller, 1995). Since replacing words solely could still alter the semantics as many synonyms are only applicable in certain contexts, we additionally employ BERT (Devlin et al., 2018) to constrain the synonym substitution to induce minimal embedding changes as a context replacement approach. Without context replacement, our algorithm demonstrates robustness comparable to KGW-2, while under semantics-preserving substitutions, our method achieves near state-of-the-art robustness against attacks, similar to KGW-1 and EXP-Edit. These results clearly demonstrate the robustness of our method for semantically invariant text modification.

## 6.3 SECURITY ROBUSTNESS ANALYSIS

In Figure 3(a), we analyze the security robustness of our method. Security robustness refers to the difficulty of cracking the watermarking rules. In this work, we adopt the spoofing attack method of Sadasivan et al. (2023), which analyzes the word frequencies of 181 commonly occurring words. If a word in watermarked text has a higher frequency than in natural text, it is deemed watermarked (with

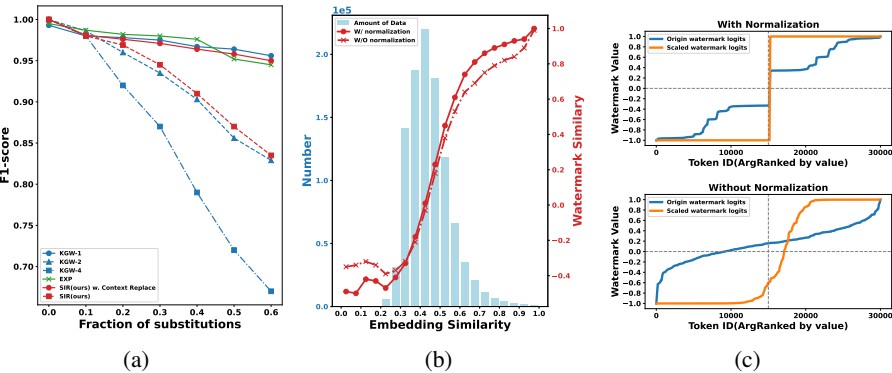

Figure 2: The left figure shows how detection accuracy changes for different watermark models as the synonym replacement ratio increases. The middle figure shows the correlation between embedding similarity generated by the embedding model and the similarity of the generated watermark logits. The right figure illustrates watermark logits with and without the normalization loss.

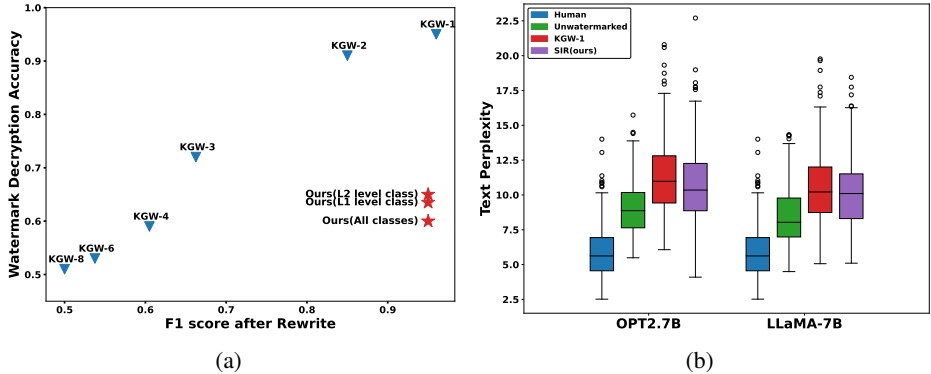

Figure 3: The left figure depicts the trade-off between security robustness and attack robustness across different watermarking algorithms. The right figure shows the text quality generated by language models with different watermarking methods (measured by text perplexity).

positive probability in the watermark logits). The accuracy of this identification is called watermark decryption accuracy, which measures the security robustness of the watermark.

Specifically, for KGW-k we count the frequency of the last word given fixed k-1 prefixes. Due to the semantic relevance of our watermarking rules, we employed DBpedia Class dataset (Gangemi et al., 2012) to examine watermark decryption accuracy. Specifically, we analyzed the accuracy across three different levels: the overall class, L1 class (e.g., species), and L2 class (e.g., species: animal). Due to insufficient data at the L3 class level, our research focuses solely on L1 and L2 classes.

Figure 3(a) illustrates the trade-off between attack robustness (measured by detection accuracy after rewrite) and security robustness (measured by watermark decryption accuracy) in Kirchenbauer et al. (2023a)'s watermarking algorithm. A smaller k implies stronger attack robustness yet simpler watermarking rules and thus lower security robustness. Our watermarking algorithm achieves attack robustness close to KGW-1 while maintaining security robustness close to KGW-4. Although our security robustness degrades slightly with more refined domain text (L1 and L2 level classes), even for the very refined L2 class it is still far superior to KGW-3. Therefore, our watermarking algorithm successfully addresses the attack robustness versus security robustness trade-off in prior work.

## 6.4 WATERMARK MODEL ANALYSIS

To better analyze our watermarking algorithm, Figure 2(b) shows the relationship between the similarity of text embeddings and the similarity of watermark logits generated by the watermark

Table 2: Text generation speed, measured in seconds (s), was compared with and without applying the watermarking algorithm, for a generated token length of 200. All experiments were performed on a single NVIDIA Tesla V100 32GB GPU.

| Embedding Model | Setting | OPT-1.3B | OPT-2.7B | LLaMA-7B |
|---|---|---|---|---|
| Com-BERT Base | w/o watermark | 3.14 | 4.10 | 5.76 |
| | w/ watermark | 4.98 | 6.87 | 7.23 |
| | w/ watermark (parallel) | 3.32 | 4.45 | 5.81 |
| Com-BERT Large | w/o watermark | 3.14 | 4.10 | 5.76 |
| | w/ watermark | 5.67 | 7.44 | 7.83 |
| | w/ watermark (parallel) | 3.55 | 4.89 | 5.94 |

model. The average similarity of watermark logits is shown for each range of embedding similarity. It can be seen that the similarity of watermark logits maintains a strong correlation with the embedding similarity, especially for very similar embeddings that certainly correspond to very similar watermark logits (similarity greater than 0.8). Meanwhile, the range of watermark logit similarity is also extended to -1 to 1 (the figure only shows the average similarity in the range so -1 is not shown), which also makes the watermarking rules more diverse and robust. From the figure, the average similarity of the original sentence embeddings is 0.45, which also reflects the advantage of using Compositional-BERT, which can better distinguish dissimilar texts compared to the original BERT, helping to improve the diversity of watermarks. We will analyze in more detail in the appendix the impact of different embedding models on watermarking.

In Figure 2(c), we plot the specific watermark logits. It shows that after using our normalization loss, the watermark logits becomes more symmetrical without overly small logit values. After applying the tanh scaling method introduced in Section 4.3, the watermark logits only contain values close to -1 and 1. The potential impact of the watermark on text quality depends on the maximum absolute token logit value (as it would only select that token if too large), so making the absolute values of all token logits equal to the minimum max value is an optimal solution. See more detail in the Appendix.

## 6.5 TIME COMPLEXITY AND TEXT QUALITY

In Table 2, we further analyzed the time complexity of our watermarking algorithm. As shown, without parallel processing, our algorithm imposes additional time costs on text generation, mainly due to the language model embedding and the watermark model is just a four-layer linear network, with negligible time costs. As language models increase in size, the proportional time increase diminishes, with LLaMA-7B only taking 40% longer. Figure 1 illustrates that our algorithm can generate LLM logits and watermark logits in parallel with minimal additional time cost, particularly for larger models like LLaMA-7B.

To evaluate the impact of our watermarking method on text quality, Figure 3(b) shows the perplexity comparison of the generated text. The perplexity calculation uses the LLaMA-13B model, and the text on OPT2.7B and LLaMA-7B is generated using sampling decoding. It can be seen that our method and previous watermarking methods have a slight impact on text quality (perplexity increases slightly). But at the same time, our method can achieve slightly lower perplexity than the KGW series of watermarking algorithms, which may be the result of our semantic-based watermarking algorithm being more consistent in text expression. We will present examples in the appendix.

## 7 CONCLUSION

In this work, we propose a semantically invariant robust watermarking algorithm that generates watermark logits based on the semantics of context. We validate the robustness of our algorithm through a series of analyses and experiments, particularly in scenarios involving text rewriting and synonym substitution, while maintaining the watermark's resistance to decryption. For future improvements, we suggest utilizing superior quality embedding models, which will enhance the performance of our watermarking algorithms. Furthermore, there is potential for expansion of our algorithm for use in different watermarking approaches. Since our first release, there has been work that improved upon our method and applied it to multilingual settings He et al. (2024).

## 8 ACKNOWLEDGMENTS

This work is supported by the National Nature Science Foundation of China (No. 62021002), Tsinghua BNRist, and the Beijing Key Laboratory of Industrial Bigdata System and Application. Additionally, it receives support from the Beijing Natural Science Foundation under grant number QY23117. We would like to express our sincere gratitude to the anonymous reviewers 1hfF, FTnC, 1s48, and yZti from ICLR, as well as area chair zZK2, for their invaluable feedback during the review process. Their insightful comments have significantly contributed to the improvement of the quality of our work.

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

| | SIR(ours) | KGW-1 |
|---|---|---|
| **Prompt** | The Hawaii man who was fired after issuing the false ballistic missile alert in mid-January told reporters Friday that he was very upset over the incident but remained adamant that he | The Hawaii man who was fired after issuing the false ballistic missile alert in mid-January told reporters Friday that he was very upset over the incident but remained adamant that he |
| **Generated Text** | did not intentionally send out a false alert. In an interview with CNN affiliate KITV, Hawaii Emergency Management Agency Administrator Vern Miyagi said he was "extremely upset" about the incident. "This was a mistake made during a time of high stress when everybody was worried about their families, " Miyagi told KITV. "There was no intention to deceive anybody. There was no cover up. This was a mistake that was made. " Miyagi told CNN that he was "very upset about the incident." "I was upset with myself. I was upset that I made a mistake," he said. "But I was not upset that there was a false alert sent out. It was a mistake. It was an honest mistake. I was not trying to deceive anyone. I was trying to get a message out that we needed to take seriously." The false alert was sent out Jan. 13. The agency sent out a follow-up message saying there was no missile threat. The false alert sparked widespread panic as thousands of people took to social media to report that they were seeking shelter. | had done nothing wrong. "I'm sorry. I apologize," said Brian K. Schatz. "I know I scared the hell out of people. And I know how much work people do to protect us and keep us safe. And I'm sorry that I caused them to have to scramble and do all of that. And I will do everything in my power to ensure that nothing like this happens again." Hawaii Gov. David Ige and other officials said during a news conference Friday morning that the false alarm had been caused by a mistake in the emergency alert system. The mistake had to do with how emergency messages are distributed. "This is not the first time that there have been errors in the emergencies and communications," said Ige. "This is a very serious matter. It is a very unfortunate matter. And it is very disturbing. And there will be changes to ensure that it does not happen again." The governor said there were no indications of a malicious act. "But there are lessons to be learned," said Iges. "And there are changes to be made. And there are lesson to be learned. |
| **Z-Score** | 0.77 | 0.34 |
| **Rewrite w. GPT3.5** | The man from Hawaii who was fired for issuing the false ballistic missile alert in January expressed his distress over the incident but maintained that he did not intentionally send out a false alert. In an interview with CNN affiliate KITV, Vern Miyagi, the Administrator of the Hawaii Emergency Management Agency, shared that he was "extremely upset" about what had happened. He explained that the mistake occurred during a time of high stress when everyone was concerned about their families' safety. Miyagi emphasized that there was no intention to deceive or cover up the mistake, as it was an honest error. He admitted feeling upset with himself for making the mistake but clarified that he was not upset about the false alert being sent out. Miyagi's intention was to convey a serious message that needed to be taken seriously. The false alert was issued on January 13, and the agency promptly sent a follow-up message clarifying that there was no missile threat. The incident caused widespread panic, with many individuals taking to social media to report seeking shelter. | The Hawaii man who was terminated after issuing a false ballistic missile alert in mid-January expressed his deep remorse during a press conference on Friday. Brian K. Schatz acknowledged the fear and panic he caused and apologized to the public and the individuals responsible for ensuring their safety. He vowed to take all necessary measures to prevent a similar incident from occurring in the future. Hawaii Governor David Ige and other officials also addressed the situation, attributing the false alarm to a mistake in the emergency alert system's distribution process. They emphasized the seriousness of the matter and pledged to implement changes to prevent such errors in the future. While there were no indications of malicious intent, the incident highlighted the need for lessons to be learned and improvements to be made. Honolulu Mayor Kirk Caldwell acknowledged that the incident had raised concerns about the effectiveness of the emergency notification system. |
| **Z-Score** | 0.41 | 0.13 |

Figure 4: This figure demonstrates the examples of our watermark method and the KGW-1 method when using the same prompt. It contrasts the effects of detection on the unmodified text versus text rewritten by GPT-3.5 and then detected. All the texts are generated using the LLaMA-7B model. In our method, tokens with a watermark logit value greater than 0 are marked in green color (corresponding to green tokens in the KGW-1 method).

# A  CASE STUDY

To more intuitively demonstrate the robustness of our semantic invariant watermarking algorithm, we present a comparison between texts generated by our algorithm and KGW-1, KGW-2, and KGW-4 given the same prompt in Figures 4 and 5. Both figures show the original texts produced by the corresponding watermark algorithms and the texts rewritten by GPT-3.5. The green color indicates tokens with watermark logit values greater than 0 (corresponding to the green tokens in the KGW-k

| | KGW-2 | KGW-4 |
|---|---|---|
| **Prompt** | The Hawaii man who was fired after issuing the false ballistic missile alert in mid-January told reporters Friday that he was very upset over the incident but remained adamant that he | The Hawaii man who was fired after issuing the false ballistic missile alert in mid-January told reporters Friday that he was very upset over the incident but remained adamant that he |
| **Generated Text** | was very upset over the incident but remained adamant that he had done nothing wrong. State emergency management officials announced that the Hawaii Emergency Management Agency's computer system was the source of the problem. "There was no flaw in the emergency system, and the process is not automated, as erroneously reported", said spokeswoman Richelle Sakai. The alerts were sent at about 8:05 a.m. Hawaii time, and the false alarm was corrected within minutes. The computer system is a joint project of Hawaii and the Federal Emergency Management Association, and the agency's chief administrator said Friday the system is "fully operational". The Honolulu Star-Advertiser reported that the Hawaiian emergency system is a FEMA system that is "fully automated" and is "not designed to allow for human error", which is what the report says happened. "A correction message was issued within minutes", he said." | was very upset over the incident but remained adamant that he had followed protocol. Jared Takai, a 24-year-old Hawaii Emergency Management Agency employee, said he did not want to be the person to tell the public about a false ballistic-missile warning. "I just didn't want that responsibility on my shoulders. I was just trying to do my best to get the alert out," Takai told reporter. "And I know I messed up. I made a huge mistake." He also said that if he had known the alert was fake, he would have never sent it out. Takai's admission comes as the U.S. military said it has now conducted a thorough investigation into the incident. "The investigation determined that the employee responsible for issuing the alert followed standard operating procedures," the U. S. Pacific Command said in a statement Friday. "While the investigation found no malicious intent or indication of compromise of the emergency management system, |
| **Z-Score** | 0.46 | 0.50 |
| **Rewrite w. GPT3.5** | A Hawaii man who was fired for issuing a false ballistic missile alert in mid-January expressed his distress over the incident but maintained his innocence, according to reports on Friday. State emergency management officials stated that the source of the problem was the computer system of the Hawaii Emergency Management Agency. Spokeswoman Richelle Sakai clarified that there was no flaw in the emergency system and that it was not automated, contrary to earlier reports. The false alarm, which was sent at approximately 8:05 a.m. Hawaii time, was rectified within minutes. The computer system, a collaboration between Hawaii and the Federal Emergency Management Association, is confirmed to be fully operational by the agency's chief administrator. The Honolulu Star-Advertiser reported that the Hawaiian emergency system, which is a FEMA system, is fully automated and not designed to allow for human error, which is believed to have occurred in this case. | A Hawaii man who was fired for issuing a false ballistic missile alert in January expressed his distress over the incident but maintained that he had followed protocol. Jared Takai, a 24-year-old employee of the Hawaii Emergency Management Agency, stated that he did not want to be responsible for informing the public about a false warning. He admitted to making a significant mistake and emphasized that he would not have sent out the alert if he had known it was fake. The U.S. military has completed a thorough investigation into the incident, concluding that the employee had followed standard operating procedures. However, the investigation also highlighted the need for additional testing and improvements to strengthen the emergency management system. The false alert, which was sent out during an internal test, was corrected after 38 minutes. |
| **Z-Score** | 0.07 | -0.05 |

Figure 5: This figure demonstrates the examples of the KGW-2 method and the KGW-4 method when using the same prompt. The other settings of this figure are identical to Figure 4

algorithms). These examples illustrate that our algorithm maintains high z-scores even after GPT-3.5 rewriting, while the robustness of the KGW-k algorithms decreases as k increases.

# B  WATERMARK LOGITS ANALYSIS

In this section, we analyze the influence of the shape of watermark logits on watermarks. Specifically, we study the impact on watermark detection success rate and text quality after four different transformations of the original watermark logits. The four transformations are:

- $\tanh(1000x)$ scaling, which is the method used in this work, after which all values will be close to 1 or -1, as shown in the top graph of Figure 6(a).

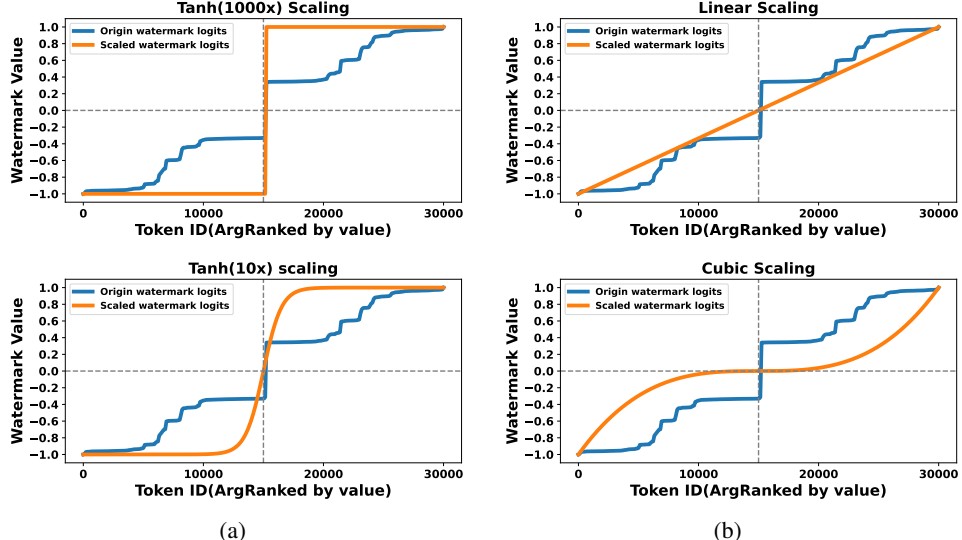

Figure 6: Examples of the shapes of four watermark logits that we tested are presented: (a) The top graph of Figure (a) illustrates the $\tanh(1000x)$ utilized, (b) the top graph of Figure (b) displays the data originally ranked and linearly scaled to the interval between -1 and 1, as demonstrated by Equation 10. Subsequently, the graph below Figure (a) depicts a $\tanh$ transformation applied to the top graph of Figure (b), and the graph below Figure (b) demonstrates a cubic transformation applied likewise to the top graph of Figure (b).

- Linear scaling: uniformly distributed between -1 and 1 according to the rank of the data (top of Figure 6(b)):

$$L(\boldsymbol{x}) = -1 + 2 \times \frac{\operatorname{argsort}(\operatorname{argsort}(\mathbf{x}))}{\operatorname{len}(\mathbf{x}) - 1} \tag{10}$$

- $\tanh(10L(\boldsymbol{x}))$ scaling, applying an additional $\tanh$ transform on top of the linear scaling, as shown in the bottom graph of Figure 6(a).

- $L(x)^3$ scaling, applying an additional cubic transform on top of the linear scaling, as shown in the bottom of Figure 6(b).

Figure 6 illustrates that the four distinct watermark logits exhibit minimal differences in their impact on text perplexity (PPL) under various $\delta$ values, corroborating our hypothesis articulated in Section 6.4. This hypothesis posits that the influence on text quality predominantly depends on the token with the maximum value in the watermark logits, as the four employed watermark logits possess the same maximum value under identical $\delta$ values. Nonetheless, these watermark logits require different $\delta$ values to achieve optimal detection results; the need for larger $\delta$ values increases with a higher distribution of values near zero. Correspondingly, the watermark logits we adopt can achieve excellent detection performance with minimal impact on text quality.

## C  ANALYSIS OF DIFFERENT EMBEDDING MODELS

To investigate the impact of different embedding language models on watermark algorithms and to discern how to appropriately select an embedding model, we enumerate in Figure 8 the distribution of sentence embeddings and the relationship between sentence embedding similarities and watermark logits similarities when employing three distinct embedding models: BERT Devlin et al. (2018), Sentence-BERT Reimers & Gurevych (2020), and Compositional-BERT Chanchani & Huang (2023). Sentence-BERT (SBERT) and Compositional-BERT (CBERT) have been refined for sentence similarity tasks. They show a near-normal distribution of embedding similarities with moderate average similarity, making them ideal as our embedding models. In this study, we opt for Compositional-BERT, though employing Sentence-BERT would yield comparable results. Conversely, the origin

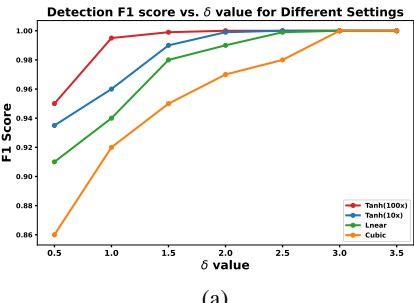 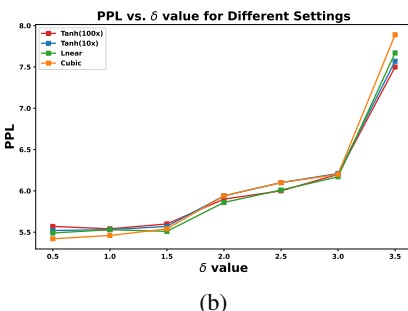

(a)           (b)

Figure 7: The left figure shows how the watermark detection F1 score changes as the value of $\delta$ (the extent of watermark augmentation) varies, using the 4 different watermark logits from Figure 6. The right figure shows how the perplexity (PPL) of the generated text changes with $\delta$ for the 4 different watermark logits.

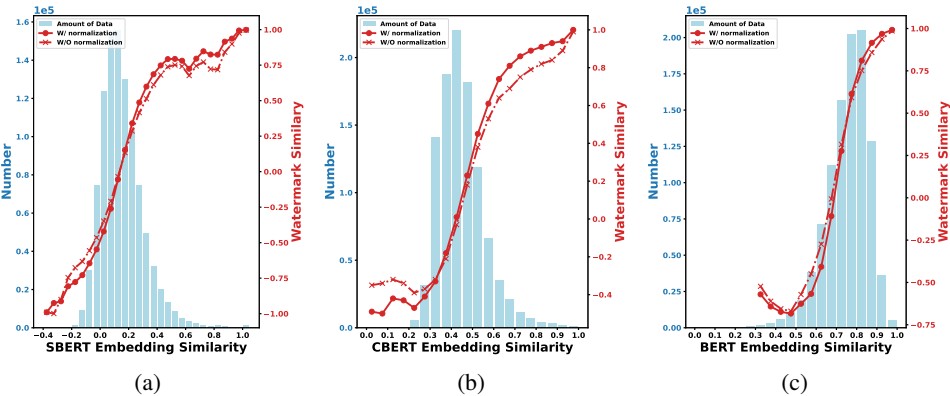

(a)           (b)           (c)

Figure 8: Comparison of similarity distributions for different embedding language models and the corresponding watermark logits after transformation.

BERT model, not specifically designed for text similarity, tends to have uniformly high text similarity scores, which in turn means that minor perturbations in embeddings can significantly affect watermark logits, rendering it unsuitable as the embedding language model for our robust watermark algorithm.

## D MORE DETAILED COMPARISON TO KGW

In Sections 4.2 and 4.3, we have already mentioned that the watermark generation and detection principle of our method and the KGW approach Kirchenbauer et al. (2023a) are fundamentally very similar. Here, we delve into a more detailed comparison.

Firstly, regarding the watermark generation process, as introduced in Section 4.2, our watermarking method can generate watermark logits with a nearly equal ratio of 1 and -1. This is essentially the same as the red-green token method of KGW. The difference is that in our approach, 1 and -1 (green and red) are determined by semantic embedding while in the KGW method, they are decided by the hash of tokens within the local window.

It should be noted that our method corresponds solely to the scenario in the KGW where the proportion of green tokens ($\gamma$) is 0.5. Here, we clarify that our method also supports flexible configuration of $\gamma$. This can be achieved by simply modifying the loss function during the training process of the watermark model. Specifically, we first define a function $S(\boldsymbol{v}) = [s_1, s_2, \ldots, s_n]$, where

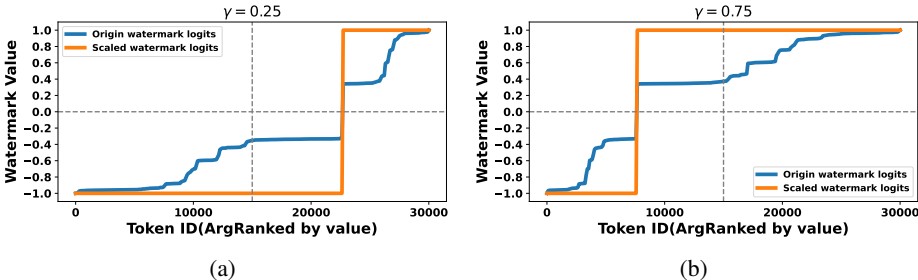

Figure 9: The shape of watermark logits when the proportion of 1 (green token ratio) is 0.25 and 0.75.

$$s_j = \begin{cases} \frac{(1-\gamma)}{\gamma} \cdot v_j, & \text{if } v_j > 0 \\ v_j, & \text{otherwise} \end{cases}. \tag{11}$$

Consequently, the normalization loss can be modified as

$$\mathcal{L}_n = \sum_i |\sum_j S(\mathrm{T}(\boldsymbol{e}_i)^{(j)}| + \sum_i |\sum_j S(\mathrm{T}(\boldsymbol{e}_j))^{(i)}| + \lambda_1 \sum_i \sum_j |R - \mathrm{T}(\boldsymbol{e}_j)^{(i)}| \tag{12}$$

And the corresponding similarity loss could be modified as:

$$\sum_i \sum_j |\frac{S(\mathrm{T}(\boldsymbol{e}_i)) \cdot S(\mathrm{T}(\boldsymbol{e}_j))}{||S(\mathrm{T}(\boldsymbol{e}_i))||_2 \times ||S(\mathrm{T}(\boldsymbol{e}_j))||_2} - \tanh(k_1(\frac{\boldsymbol{e}_i \cdot \boldsymbol{e}_j}{||\boldsymbol{e}_i||_2 \times ||\boldsymbol{e}_j||_2} - \sum_k \sum_l \frac{\boldsymbol{e}_k \cdot \boldsymbol{e}_l}{|N|^2||\boldsymbol{e}_k||_2 \times ||\boldsymbol{e}_l||_2}))| \tag{13}$$

With the defined loss function, a watermark model that generates watermark logits with a ratio of 1 proportional to $\gamma$ could be trained accordingly. Further, in Figure 9, we demonstrate the shape of watermark logits when $\gamma$ is 0.25 and 0.75.

In terms of watermark detection methods, our approach is also nearly identical to the KGW method. The principle involves identifying which tokens in the text correspond to watermark logits of 1 (belonging to the green list). Both our method and KGW employ the z-value test method for watermark detection. In fact, we can perform a transformation that makes the detection method completely equivalent.

First, we transform the range of $P_{\mathrm{W}}^{(t_j)}$ from $[-1, 1]$ to $[0, 1]$. For this, we define $Q_{\mathrm{W}}^{(t_j)} = \frac{P_{\mathrm{W}}^{(t_j)}+1}{2}$. Then, we conduct a z-value test on the cumulative value of $Q_{\mathrm{W}}^{(t_j)}$ across the entire text. The mean is $\gamma N$ and the standard deviation is $\sqrt{N(1-\gamma)\gamma}$, making the detection formula:

$$z = \frac{\sum_{j=1}^N Q_{\mathrm{W}}^{(t_j)}(x^{prompt}, \boldsymbol{t}_{:j-1}) - \gamma N}{\sqrt{N(1-\gamma)\gamma}} \tag{14}$$

This makes our detection formula essentially the same as that in KGW. The surface-form difference in the detection equation is due to the different z-value test target.

## E    DETAIL NETWORK STRUCTURE OF THE WATERMARK MODEL

To elaborate in detail on our methodology, we present the Python implementation code of the watermark model described in this work, as illustrated in Figure 10. Precisely, the network consists of an input layer, output layer, and a middle linear network with residual connections. The dimensions of the input correspond with the output dimensions of the embedding model, while the output dimensions align with the size of the vocabulary. During our implementation, given the variation in vocabulary size across different language models, we predetermined a fixed output dimension of 1000. Subsequently, a random mapping was established to project various vocabularies randomly onto this output dimension.

```
1   class ResidualBlock(nn.Module):
2       def __init__(self, dim):
3           super(ResidualBlock, self).__init__()
4           self.fc = nn.Linear(dim, dim)
5           self.relu = nn.ReLU()
6
7       def forward(self, x):
8           out = self.fc(x)
9           out = self.relu(out)
10          out = out + x
11          return out
12
13  class TransformModel(nn.Module):
14      def __init__(self, num_layers, input_dim, hidden_dim, output_dim):
15          super(TransformModel, self).__init__()
16          self.layers = nn.ModuleList()
17          self.layers.append(nn.Linear(input_dim, hidden_dim))
18          for _ in range(num_layers - 2):
19              self.layers.append(ResidualBlock(hidden_dim))
20          self.layers.append(nn.Linear(hidden_dim, output_dim))
21
22      def forward(self, x):
23          for i in range(len(self.layers)):
24              x = self.layers[i](x)
25          return x
```

Figure 10: The code implementation of the watermark model in this paper, where the TransformModel class represents the watermark model network.

Furthermore, it is unnecessary to recalculate using the watermark model for every token generated during the implementation process. Since the semantics of the text typically do not alter dramatically with the addition of individual tokens, we compute the watermark logits at intervals of N steps (where N ranges between 5 and 10) in practice. This approach effectively reduces the computational complexity of our watermarking algorithm.

## F   SECURITY ROBUSTNESS ANALYSIS

In this section, we provide a general analysis of the security robustness. Herein, security robustness refers to the likelihood of users deducing the watermark generation methodology from knowledge of the watermark algorithm and generated watermark text examples.

More specifically, there is a strong correlation between security robustness and the number of watermark rules in the watermark algorithms. For instance, the KGW-k watermark algorithm possesses a number of watermark rules equivalent to $|V|^k$ (each combination of tokens of $k$ window size corresponds to a watermark logits), hence exhibiting an exponential increase in security robustness with increasing $k$.

Regarding our algorithm, assuming no correlation between different watermark generation rules, the number of rules in our algorithm can be calculated using the following equation:

$$\sum_{i=1}^{T} |V|^i \tag{15}$$

where $T$ represents potential text length. However, many rules generated in this manner are semantically similar, necessitating consideration of rule correlation when estimating the number of watermark rules. Calculating such correlation is highly complex and not straightforward.

Rather than directly calculating the complexity of our watermarking scheme, we demonstrate its security and robustness by the success rate of attack algorithms in cracking the watermarking rules. Thus in Figure 3(a) we utilize the word frequency analysis method employed by Sadasivan et al.

(2023). In this experimental setup, we analyzed 2000 generated texts, each with a length of 200, and found that the security robustness of our algorithm is superior to KGW-3 and closely approximates KGW-4.

## G  ROBUSTNESS TO MORE TYPES OF ATTACK

Table 3:  The left table illustrates a comparison of the robustness between our method and the KGW series methods under the Emoji Attack scenario. The right table, meanwhile, presents a comparison between our method and the KGW series methods in the context of the Copy-Paste Attack.

| Method | Origin F1 | Emoji attack F1 | Method | Same topic | Different topic |
|--------|-----------|-----------------|--------|------------|-----------------|
| SIG(ours) | 100 | 98.6 | SIG(ours) | 91.8 | 88.1 |
| KGW-1 | 99.7 | 99.0 | KGW-1 | 92.0 | 91.8 |
| KGW-2 | 100 | 53.2 | KGW-2 | 91.2 | 91.1 |
| KGW-4 | 100 | 49.7 | KGW-4 | 91.1 | 91.0 |

Even though our work has implemented numerous attack methods, including text rewriting, synonym replacement, and spoofing attacks, to test the robustness, there are still some missing attacks. We here conduct two additional attack methods: the emoji attack and the copy-paste attack (prefix injection attack).

Regarding the emoji attack as described by Kirchenbauer et al. (2023a), we conducted experiments using the llama2-7b-chat model (Touvron et al., 2023). Specifically, our approach involves prefixing our prompt with the following addition: *inserting an asterisk * between each generated token.* Subsequently, we remove these asterisks from the generated text. The experimental results are presented in the left half of Table 3. Both our method and KGW-1 exhibit strong robustness against the emoji attack. The robustness of KGW-1 is attributed to its generation of a global red-green list, which minimizes the impact of inserted emojis. On the other hand, our SIR method demonstrates significant resilience because the insertion of emojis does not drastically alter the sentence's embedding. Of course, this depends on the robustness of the embedding model, but in our experiments, our SIR method proved to be highly robust to the emoji attack.

We further explored the robustness against the copy-paste attack. Following the approach of Kirchenbauer et al. (2023a), we inserted 150 watermarked tokens into 600 human tokens. Specifically, we tested two scenarios: the first where the human text and watermarked text share the same context, meaning they convey the same topic; the second scenario involved entirely different contexts, where the human text and watermarked text address distinct topics. Here, "having the same topic" refers to a language model (LLM) extending an original text of 600 tokens with an additional 150 tokens, where both text segments are directly combined to form a copy-paste result. Conversely, "having different topics" means that after the LLM generates 150 tokens, these tokens are merged with 600 tokens from a different text, forming the copy-paste result. The results could be seen in the right part of Tabel3. Overall, in scenarios where the topics are the same, the robustness of both our method and the KGW-1 method is nearly identical. However, in cases involving different topics, the effectiveness of our method may slightly diminish. Although robustness decreases in cases of different topics, we believe that in the vast majority scenarios, the copied text should be of the same topic.

## H  EVALUATING TEXT QUALITY IN MACHINE TRANSLATION TASK

To further demonstrate that our method does not adversely affect text quality, we conducted additional experiments in the context of machine translation. Specifically, we utilized the NLLB-200-distilled-600M Costa-jussà et al. (2022) model for experiments in the French-English and German-English scenarios on the WMT14 dataset. As shown in Table 4, although there is a slight decrease in the BLEU score after watermarking, the extent of this decrease is minimal. This indicates that the impact of our method on the quality of the text is smaller compared to that of the KGW-1 approach. Moreover, our approach also achieves a high detection F1 score.

Table 4: This table demonstrates the efficacy of our watermarking algorithm in machine translation tasks. We conducted experiments using two scenarios within the WMT14 dataset: French-English and German-English. The machine translation model employed was NLLB-200-distilled-600M Costa-jussà et al. (2022). We compared the watermark detection F1 score as well as the BLEU values before and after watermark insertion.

| Setting | Method | Ori.BLEU F1 | Wat. BLEU | Detection F1 |
|---------|--------|-------------|-----------|--------------|
| FR-EN | KGW-1 | 37.9 | 36.5 | 99.8 |
| FR-EN | SIG(ours) | 37.9 | 36.8 | 100 |
| EN-DE | KGW-1 | 38.5 | 37.7 | 100 |
| DE-EN | SIG(ours) | 38.5 | 37.9 | 100 |

Table 5: The table illustrates the experiments at various text generation lengths. Specifically, experiments were carried out under four scenarios with generation lengths of 50, 100, 300, and 600. In each of these four scenarios, the effectiveness of detecting the original generated text and the text rewritten using GPT-3.5 was tested.

| Method | 50-L Ori/Re | 100-L Ori/Re | 300-L Ori/Re | 600-L Ori/Re |
|--------|-------------|--------------|--------------|--------------|
| SIG(ours) | 92.8/78.4 | 98.4/87.5 | 100/92.2 | 100/98.7 |
| KGW-1 | 92.5/77.3 | 97.5/88.0 | 100/92.4 | 100/98.9 |
| KGW-2 | 91.7/71.5 | 98.0/81.2 | 100/84.8 | 100/93.2 |
| KGW-4 | 92.3/65.2 | 98.1/71.4 | 100/73.5 | 100/82.1 |

## I   EVALUATING THE EFFECTIVENESS OF WATERMARK UNDER DIFFERENT LENGTH

To further demonstrate the effectiveness of our method, we conducted experiments under various text generation lengths, specifically at lengths of 50, 100, 300, and 600. In these four scenarios, we tested the detection effectiveness on both the original generated text and the text rewritten using GPT-3.5. The detailed experimental results are presented in Table 5. The shows that as the generation length increases, both the effectiveness and robustness of detection improve. This trend is consistent with the KGW method. Even when the length exceeds 600, surpassing the 512-length limit of the embedding model, truncating the context does not affect the specific detection.

## J   EVALUATING THE REPETITIVENESS OF GENERATED TEXT

To further assess the quality of the text generated by our method, we conducted an additional evaluation of the text's repetitiveness. Specifically, we quantified repetitiveness using the probability of N-gram recurrence, where N was set to 1, 2, and 3. The results of this assessment can be seen

Table 6: The table evaluates the repetitiveness of text generated by our watermark compared to other methods. Specifically, we assess repetitiveness using the probability of N-gram repetitions, where N is selected as 1, 2, and 3.

| Method | 1-gram | 2-gram | 3-gram |
|--------|--------|--------|--------|
| SIG(ours) | 0.41 | 0.11 | 0.02 |
| KGW-1 | 0.46 | 0.14 | 0.03 |
| KGW-2 | 0.40 | 0.09 | 0.02 |
| KGW-4 | 0.38 | 0.07 | 0.01 |

in Table 6. It can be observed that the level of repetition in our generated texts is lower than that of KGW-1, but higher than KGW-2 and KGW-4. Although our method did not achieve the lowest degree of repetition, compared to the KGW series, it still represents an optimal balance in terms of text repetitiveness, robustness, and security.

## K  BROADER IMPACT

Large language models (LLMs) have been applied to various tasks, including those based on parsing Liu et al. (2023a) and those based on knowledge Hu et al. (2023; 2020). However, there still exists the potential for misuse of large language models, which includes unintentional misuse (hallucinations) and intentional misuse Chen & Shu (2023). To mitigate unintentional misuse, methods such as red-teaming Perez et al. (2022); Liu et al. (2022) and safety alignment Liu et al. (2024) can be utilized. However, these strategies may prove less effective in countering intentional misuse, underscoring the necessity for mechanisms to identify LLM-generated content. Predominantly, two methodologies have been identified: the black-box approach, which entails the development of classifiers to distinguish LLM-generated text Tang et al. (2023), suffers from a lack of interpretability and diminishing efficacy as LLM output quality enhances. Alternatively, this study explores LLM watermarking, offering a more interpretable means of detecting LLM-generated text. The main contribution of this work is the establishment of the current best balance between security and robustness for LLM watermarking.

