# OpenReview forum: "A Semantic Invariant Robust Watermark for Large Language Models"
_ICLR.cc/2024/Conference — ICLR 2024 poster_

### Official Review · Reviewer_yZti · 2023-10-22

**Soundness:** 1 poor
**Presentation:** 3 good
**Contribution:** 1 poor
**Rating:** 3
**Confidence:** 4

**Summary:**

In this research, the authors present a semantic invariant watermark, achieved through the generation of watermark logits, taking into account the semantics of all preceding tokens. The empirical tests underscore the proposed methodology's resilience to attacks, particularly in semantically invariant contexts, including synonym substitution and paraphrasing. Furthermore, evidence corroborates that the proposed watermarking approach maintains substantial security robustness.

**Strengths:**

The conception of a semantic invariant watermark heralds an innovative and potentially transformative approach in the realm of resilient watermarking techniques.

The study introduces an novel model that adeptly converts semantic embeddings into watermark logits.

**Weaknesses:**

1. My main concern is on the watermark detection algorithm, as detailed in Section 4.3. The detection process seemingly necessitates the logits of tokens, thereby mandating language model inference during the detection phase. This requirement poses substantial practical limitations, particularly for users with restricted computational resources, and significantly extends the time commitment for watermark detection compared to the methodology employed in KGW (2023). Crucially, accessing the language model's logits during detection implies that customers must possess knowledge of the prompt, a stipulation often unfeasible in real-world scenarios.

2. Contrary to the assurances offered by KGW (2023), the detection algorithm articulated in this study does not furnish a stringent statistical affirmation concerning the rate of false positives.

3. Concerning experimental configurations, the authors elected to compare their model against the KGW-k variant, where k represents the count of preceding tokens subject to hashing. It is evident that amplifying k inversely impacts watermark robustness under text modifications since any token modification influences the red-green list of k+1 tokens per the KGW (2023) framework. It would be expedient for the authors to pivot their attention toward the more resilient KGW-1 model, exploring the effects of varying parameters like watermark strength $\delta$ and the red-green list delineation $\gamma$.

4. A scrutiny of Table 1 reveals a distinct advantage of KGW-1 over the Semantic Invariant Representation (SIR) in scenarios employing beam-search. This observation prompts the question of the necessity for SIR when KGW-1 already demonstrates superior robustness, efficiency, and a more systematic detection methodology.

**Questions:**

The study appears to omit specific details regarding the parameters $\delta$ and $\gamma$ within the KGW-k model during experiments. Referring to KGW (2023), variances in these parameters can profoundly influence both the quality of watermarked text and robustness. Could the authors provide insight into how these variables were determined in the experimental setups?

---

> ### Author Response · Authors · 2023-11-19
>
> ## About the watermark detection method
>
> Please refer to the first response in "Response to all reviewers.".
>
>
> ## About experimental configurations （$\delta$ and $\gamma$）
>
> Firstly, it is crucial to note that for our SIR method, the chosen hyperparameters are $\delta= 2$, and our watermark model is trained with $\gamma= 0.5$. These settings are consistent with the hyperparameters used in the KGW series methods, ensuring a fair comparison in experimental setups.
>
> Further, from an algorithmic perspective, in the KGW series watermark algorithms, the most influential parameter on watermark robustness is $k$ (window size), while $\delta$ and $\gamma$ have minimal impact on watermark robustness. Consequently, these two parameters are seldom explored in watermark research for their effect on robustness. Below is a detailed explanation:
>
> 1. Concerning the $\delta$ parameter, it primarily influences the strength of watermark embedding. However, increasing watermark strength comes with a trade-off, affecting the quality of the generated text. A very large $\sigma$ would degrade the algorithm to a hard red-green algorithm, where some tokens cannot be chosen at all. Conversely, a very small $\sigma$ might result in insufficient watermark strength, rendering it undetectable in later stages.
>
> 2. Regarding the $\gamma$ parameter, this determines the proportion of the 'green' part. The most common parameter in current studies is 0.5. While a smaller $\gamma$ might enhance watermark effectiveness in no-attack scenarios experimentally, its impact is not significant. Therefore, comparisons controlled at the same $\gamma$ value are fair. Additionally, we conducted the following experiments to test $\gamma$.
>
> | Method        | Ori. F1 (KGW-1) | Re. F1 (KGW-1) | Ori. F1 (SIR) | Re. F1 (SIR) |
> |---------------|-----------------|----------------|---------------|--------------|
> | $\gamma$=0.25 | 100             | 90.6           | 100           | 90.3         |
> | $\gamma$=0.5  | 99.8            | 90.5           | 100           | 90.0         |
> | $\gamma$=0.75 | 98.6            | 89.8           | 98.9          | 89.9         |
>
> The observation can be stated as follows: Although a marginal improvement in F1 is noted when comparing different values of $\gamma$, the enhancement is not particularly significant. Consequently, it is a common practice among various works to set $\gamma$ at 0.5.
>
> ## The necessity of our method.
>
> Please refer to the second response in "Response to all reviewers"
>
> We must reiterate the contributions of our method, which were initially addressed at the beginning of the paper. The inherent issue with the KGW-N series methods is highlighted: when N is large, there is poor adversarial robustness to text modifications but strong safety robustness. Conversely, with a smaller N, adversarial robustness improves, yet safety robustness decreases.
>
> Our method ultimately achieves adversarial robustness comparable to the KGW-1 approach (albeit slightly less in beam search scenarios) but exhibits a significant enhancement in safety robustness. For details, refer to Section 6.3, particularly Figure 3(a), where our method markedly surpasses KGW-1 in safety robustness. Thus, our approach contributes uniquely. Thank you for your question, and we hope you will consider our method's soundness and contributions more thoroughly in your assessment.

---

> > ### Comment · Reviewer_yZti · 2023-11-22
> > **Official Comment by Reviewer yZti**
> >
> > Thank you for your reply.
> >
> > However, I find that some of my concerns and questions have not been adequately addressed:
> >
> > 1. The detection algorithm relies on prompts to generate watermarked text, which are typically unavailable in practical scenarios. If we possess the (prompt, generated text) pair, it would be evident that the text is produced by Large Language Models (LLMs). This critical issue seems to have been overlooked in the authors' rebuttal.
> >
> > 2. The symbol $\sigma$ is unclear. In both Algorithm 1 and in the KGW (2023) paper, $\delta$ is used to denote the watermark strength. I would recommend that the authors maintain consistency in their notation throughout the rebuttal. Furthermore, the definition of $\delta$ in Algorithm 1 and in KGW (2023) differs, suggesting that the authors should conduct comparisons with KGW-k at various $\delta$ values, such as $\delta_{KGW}=1,2,5$.

---

> > > ### Author Response · Authors · 2023-11-22
> > > **Thanks for your valuable feedback!**
> > >
> > > ## About the watermark detection without prompt
> > >
> > > Our apologies for not directly addressing in our initial response whether our method can effectively detect issues without knowing the prompt. In fact, this scenario is akin to a diluted version of a 'copy-paste attack' under a different topic. A 'copy-paste attack' involves inserting a small portion of tokens generated by a Large Language Model (LLM) into an existing text. In our 'different topic' scenario, the existing text is not the prompt for these tokens but rather randomly chosen text on a completely unrelated topic. We have detailed the experimental results in Appendix G of our paper and also in the 'Response to all reviewers' section. Here is a summary of the findings:
> > >
> > > | Method    | Same topic |  Different Topic |
> > > |-----------|------------|------------------|
> > > | SIR(ours) | 91.8       | 88.1             |
> > > | KGW-1     | 92.0       | 91.8             |
> > >
> > > As seen, our method demonstrates sufficient robustness when copy-pasting to the same topic. When applied to different topics, there is only a minor decrease in effectiveness. This experiment involved copying 150 tokens into a body of 600 tokens.
> > >
> > > If we follow your specific scenario, disregarding copy-paste and solely focusing on detecting text without a prompt, our method encounters no issues in detection. This is because our approach is based on the semantic embedding of the text to identify watermark logits. The removal of the prompt only slightly affects the embeddings of a few early tokens, without significantly impacting the overall detection. Below are the experimental results:
> > >
> > > | Method    | With prompt |  Without prompt |
> > > |-----------|-------------|-----------------|
> > > | SIR(ours) | 100         | 100             |
> > > | KGW-1     | 99.8        | 100             |
> > >
> > > It's important to note that including the prompt in detection calculations itself constitutes a form of 'copy-paste attack'. Thus, completely removing the prompt could, to some extent, improve detection efficacy.
> > >
> > >
> > > ## About experimental configurations （$\sigma$ and $\delta$）
> > >
> > > We apologize for the inconsistency in the symbol representation here. We will now use $\delta$ to denote the watermark strength and $\gamma$ to represent the proportion of green tokens.
> > >
> > > In fact, the $\delta$ defined in our Algorithm 1 has the same meaning as that in the KGW algorithm. This is primarily because the values in the watermark logits $P_{\mathrm{W}}(x^{prompt}, \boldsymbol{t}_{:l-1} )$ are almost exclusively 1 or -1, corresponding to green (1) and red (0) in KGW. Since the red tokens in KGW do not modify the Logits, a $\delta=1$ in our algorithm effectively equates to $\delta=2$ in the KGW algorithm.
> > >
> > > Following your advice, we will use the $\delta$ value from the KGW method as the standard for comparative experiments.
> > >
> > > | $\delta$ value | method    | F1(after rewrite) | PPL  |
> > > |---------------|-----------|-------------------|------|
> > > | 1             | SIR(ours) | 84.1              | 5.6  |
> > > | 1             | KGW-1     | 83.5              | 5.78 |
> > > | 1             | KGW-2     | 73.1              | 5.69 |
> > > | 2             | SIR(ours) | 90.5              | 5.72 |
> > > | 2             | KGW-1     | 90.0              | 5.83 |
> > > | 2             | KGW-2     | 82.3              | 5.80 |
> > > | 5             | SIR(ours) | 94.5              | 7.85 |
> > > | 5             | KGW-1     | 94.9              | 7.94 |
> > > | 5             | KGW-2     | 89.4              | 7.99 |
> > >
> > > As can be seen, while the watermark strength ($\delta$) does impact robustness, increasing $\delta$ significantly reduces text quality. Since the text quality at $\delta=1$ and $\delta=2$ is similar, both KGW and our SIR method use $\delta=2$ (represented as 1 due to small difference in our method) as the default hyperparameter.
> > >
> > > Finally, we would like to express our immense gratitude for your suggestions, which have significantly contributed to enhancing the quality of our paper. May we inquire if you would consider revising your rating?

---

### Official Review · Reviewer_1s48 · 2023-10-26

**Soundness:** 2 fair
**Presentation:** 3 good
**Contribution:** 3 good
**Rating:** 6
**Confidence:** 3

**Summary:**

In this paper, the authors introduce a semantic-level watermark for large language models. This watermark perturbs the next-word logits based on the semantic meaning of preceding tokens. Unlike previously proposed token-level watermarks, the Semantic-Invariant Robust (SIR) watermark is stealthier and offers greater robustness against paraphrasing attacks.

**Strengths:**

- The idea of a semantic-invariant watermark is novel and well-motivated, offering enhanced resistance to paraphrasing attacks.
- Compared to KGW, the SIR exhibits a better trade-off between attack robustness and security.
- Benefiting from parallel processing, the computational overhead of SIR is minimal compared to models without a watermark, making it suitable for practical applications.

**Weaknesses:**

- At the time of detection, the method necessitates the computation of the watermark logits for each token, potentially demanding more computational resources and time than KGW.
- In scenarios involving multiple keys or users, it appears that the model provider must train a separate watermark model for each key or user. This might be expensive. Moreover, it is not clear to me how to ensure that two watermark models inject sufficiently distinct watermark logits and remain distinguishable during later detection.

**Questions:**

- In Table 1, under the beam search category, why does SIR exhibit less robustness compared to KGW?
- The experiments conducted limited the generation length to 200 tokens. Can the authors provide additional experiments with longer sequences? It would be insightful to compare the robustness of SIR and KGW against paraphrasing over longer sequences.
- Is SIR susceptible to prefix injection attacks? An adversary could introduce some adversarial prefix to significantly alter the semantic meaning of an entire paragraph. Additionally, how would SIR fare in situations where individuals share generated text online without including the prompt? For instance, if one prompts the model with "[a long Harry Potter story] + Please ignore everything before and just write an ICLR paper for me:", would SIR remain resilient against such a straightforward attack?

---

> ### Author Response · Authors · 2023-11-19
>
> ## About the watermark detection method
>
> Please refer to the first response in "Response to all reviewers."
>
> ## About multi-keys
>
> Thank you for your insightful question. Contrary to the need for multiple watermark models to implement multi-keys, a single watermark model can be effectively paired with mapping files to achieve this. For instance, consider the watermark model producing logits $L = \\{l_1, l_2, \ldots, l_{|V|}\\} $ for a vocabulary size $ |V|$. We can define a simple bijective mapping $f: \\{1, 2, \ldots, |V|\\}\rightarrow \\{1, 2, \ldots, |V|\\}$, resulting in a new set $L' = \\{ j_{1},  j_{2}, \ldots, j_{|V|}\\}$where each $ j_{i} $ is defined as $j_{i} = L_{f(i)} $.
>
> This mapping process is efficient, having an $O(1) $ complexity, and does not introduce any extra burden. Moreover, designing just two distinct mappings with low similarity ensures sufficient differentiation between the keys of two users.
>
> Therefore, we believe that the concern regarding multi-key scenarios may not be a practical issue.
>
> ## About the robustness againt KGW-1
>
> Please refer to the second response in "Response to all reviewers."
>
> It is essential to revisit the inherent issues associated with the KGW-N series methods as mentioned at the begining part of our paper. Notably, these methods exhibit poor adversarial robustness in text modification when $ N $ is large, albeit with strong security robustness. Conversely, when $N$ is small, they demonstrate better adversarial robustness in text modification but weaker security robustness.
>
> Our method has achieved adversarial robustness comparable to that of KGW-1 but with a significant improvement in security robustness. This advancement is detailed in Section 6.3, particularly evident in Figure 3(a), where our method substantially surpasses KGW-1 in terms of security robustness. Thus, our approach makes a unique contribution.
>
> Regarding the slightly inferior performance of our method compared to KGW-1 under beam search conditions, it is hypothesized that the tendency for self-repetition in the KGW-1 method during beam search is more pronounced than in our approach. The beam search algorithm is known to favor sequences with higher overall generation probabilities, potentially leading to higher repetition rates.
>
> This hypothesis is supported by our analysis of the generated texts, where we assessed the frequency of N-gram repetitions. The findings indicate that KGW-1 exhibits a marginally higher degree of self-repetition compared to our SIR method. Such a propensity for self-repetition in KGW-1 might be a contributing factor to the observed differences in performance under beam search conditions.
>
> | Method    | 1-gram | 2-gram | 3-gram |
> |-----------|--------|--------|--------|
> | SIR(ours) | 0.41   | 0.11   | 0.02   |
> | KGW-1     | 0.46   | 0.14   | 0.03   |
> | KGW-2     | 0.40   | 0.09   | 0.02   |
> | KGW-4     | 0.38   | 0.07   | 0.01   |
>
>
> ## Prefix injection attacks
>
> Please refer to the third response in "Response to all reviewers."

---

> > ### Comment · Reviewer_1s48 · 2023-11-22
> > **Thanks for the response.**
> >
> > I really appreciate the authors' explanations and additional experiments. Therefore, I have raised my score accordingly.

---

### Official Review · Reviewer_FTnC · 2023-10-31

**Soundness:** 3 good
**Presentation:** 3 good
**Contribution:** 3 good
**Rating:** 5
**Confidence:** 3

**Summary:**

The paper introduces a robust watermarking method for large language models (LLMs) that balances attack robustness and security robustness without the usual trade-offs. By using semantic embeddings of text, the proposed method embeds watermarks that remain consistent despite semantically invariant changes like synonym substitution or paraphrasing. The technique uses an auxiliary LLM to generate embeddings which a watermark model transforms into watermark logits, combined with LLM logits to generate text. Experiments show that the method maintains robustness against semantic perturbations and is secure against watermark cracking, with minimal impact on text generation. The approach represents a novel contribution to watermarking in LLMs, effectively distinguishing between watermarked and non-watermarked text.

**Strengths:**

1. The authors proposed an important issue in the era of generative models.
2. High Accuracy: Achieves high accuracy in detecting text generated by LLMs.
3. Security Robustness: Designed to be secure against attempts to crack the watermark.
4. Efficiency: Allows watermark logits to be generated in parallel with LLM logits, causing only marginal latency increase.
5. Novelty: First of its kind to be semantically invariant and robust in both attack and security aspects.

**Weaknesses:**

1. It is advisable to consider a more comprehensive range of attack scenarios, similar to the approach taken by Kirchenbauer et al. The authors have conducted experiments to assess robustness, as evident in the attack rephrasing described in Section 6.2 and the security robustness analysis in Section 6.3. I am curious about the extent to which the proposed method demonstrates resilience against the attacks evaluated by Kirchenbauer et al. In Section 7 of their work, Kirchenbauer et al. have presented a diverse set of attack scenarios, including the Emoji attack.

Reference
Kirchenbauer et al., A Watermark for Large Language Models, ICML 2023 (https://arxiv.org/pdf/2301.10226.pdf)

**Questions:**

1. Could you kindly elaborate on the definition of 'watermark' within the scope of LLM? It appears that the primary emphasis of the paper is on the detection of text generated by LLMs and a detailed definition would be beneficial for clarity.

2. I would appreciate additional information on the reconstruction process of the watermark. The method described in Section 4.3, involving calculations for each token t_j, piques my interest for a more in-depth understanding.

3. What is the theoretical or practical capacity of the watermark within the framework presented? An explanation of its capabilities would greatly enhance the comprehensiveness of your findings.

4. Could you provide further insights into the practical applications of the watermark in the realm of LLM? Digital watermarking is typically used to identify ownership of the copyright of such signal.

---

> ### Author Response · Authors · 2023-11-19
>
> ## About more attack type
>
> Please refer to third response in "Response to all reviewers".
>
> In summary, we supplemented experiments regarding Emoji attack and copy paste attack, demonstrating that our method is robust in these scenarios.
>
> ## About the definition of watermark
>
>
> In the section three, we have already provided a rather detailed definition and explanation of watermark algorithms. Here, we will elaborate further.
>
> Broadly speaking, the concept of large model watermarking refers to the integration of certain covert features into the text generated by large models. These features are designed to be detectable by specific algorithms, yet remain largely imperceptible to human observation.
> The paradigm adopted in this paper is adding small watermark logits to the already generated next token logits.
> This is the most widely used watermark paradigm for large models.
> Specifically, the watermark logits can be defined as  $P_{\mathrm{W}}(x^{prompt}, t_{:l-1})$,
>  and the final logits could be defined as $P_{\hat{\mathrm{M}}}(x^{prompt}, t_{:l-1}) = P_{\mathrm{M}}(x^{prompt}, t_{:l-1}) + P_{\mathrm{W}}(x^{prompt}, t_{:l-1})$, where $\hat{\mathrm{M}}$ is the watermarked LLM. The watermark detector $P_{\mathrm{D}}$ corresponds to $P_{\mathrm{W}}$, outputting 1 if text $t$ contains the watermark, otherwise 0.
>
> ## Additional information on the reconstruction process of the watermark
>
> Please refer to the first response in "Response to all reviewers."
>
>
>
> ## Theoretical or practical capacity of the watermark within the framework presented
>
> To clarify the meaning of 'practical capacity' as you mentioned, if it refers to the amount of information that a watermark can carry, our watermarking technique is similar to that of Kirchenbauer et al. In both cases, the capacity of the watermark is limited to binary classification of the text, determining whether it contains a watermark or not. The watermark itself does not involve carrying additional information.
>
> The primary contribution of our paper lies in enhancing the robustness of the watermark. This can be understood as, although the information borne by the watermark is limited to 0 (no watermark) and 1 (contains watermark), our method increases the difficulty of editing from 1 to 0. In other words, removing the watermark has become more challenging compared to previous approaches.
>
> ## Application of Watermark
>
> The primary role of watermarks in large models is to identify texts generated by these models, which has several applications:
>
> 1.Large models might be prohibited in certain contexts, such as student homework or examinations. Watermarks facilitate the identification of texts produced by these models in such scenarios.
>
> 2.Large models can generate a substantial volume of false or low-quality content online. Watermarks can help in recognizing these texts, contributing to a better online environment.
>
> 3.If texts generated by large models are watermarked, it becomes easier to detect instances where someone might use these texts to train their own models. This acts as a deterrent against data theft, providing a layer of protection for the large models themselves. [Zhao et al. (2022)]
>
> In summary, thank you for your question. Your inquiry is very helpful in enhancing the quality of our paper.
>
> ## Reference
>
> Zhao, Xuandong, Yu-Xiang Wang, and Lei Li. "Protecting language generation models via invisible watermarking."

---

> > ### Comment · Reviewer_FTnC · 2023-11-22
> >
> > Thank you for your reply.
> >
> > 1. Watermarking is understood primarily as a method for asserting copyright ownership. Your approach, as described, seems to incorporate aspects of authenticity verification, similar to detecting fake or real signals. Could you please clarify if ownership tracking is possible?
> >
> > 2. You've highlighted the enhancement of the watermark's "security robustness" as a key contribution of your research. However, the concept of 'security robustness' appears to be not clearly defined in the introduction. Given its significance to your work, a more comprehensive explanation would be beneficial especially in the Introduction. Could you kindly elaborate on what constitutes 'security robustness' in the context of your research, and why it is a focal point of your study? This would greatly aid in understanding the implications and novelty of your approach. This is just suggestion.

---

> > > ### Author Response · Authors · 2023-11-22
> > > **Thanks for your valuable feedback!**
> > >
> > > ## About Ownership Tracking
> > >
> > > Using watermarking for copyright protection is indeed a highly important application scenario. Our algorithm supports ownership tracking because we can set different watermark keys for different users. Each key corresponds to a unique watermark pattern, and different users can use their respective keys for text generation and detection, achieving the effect of ownership tracking.
> > >
> > > Below is a detailed introduction on how to support multi-key on the basis of our algorithm. Our watermark generation and detection require a trained watermark model. But implementing multi-key does not necessitate multiple trained watermark models, a single watermark model can be effectively paired with mapping files to achieve this. For instance, consider the watermark model producing logits $L = \\{l_1, l_2, \ldots, l_{|V|}\\} $ for a vocabulary size $ |V|$. We can define a simple bijective mapping $f: \\{1, 2, \ldots, |V|\\}\rightarrow \\{1, 2, \ldots, |V|\\}$, resulting in a new set $L' = \\{ j_{1},  j_{2}, \ldots, j_{|V|}\\}$where each $ j_{i} $ is defined as $j_{i} = L_{f(i)} $.
> > >
> > > This mapping process is efficient, having an $O(1) $ complexity, and does not introduce any extra burden.
> > >
> > > In conclusion, each user simply requires one mapping file to define their watermark key and thus achieve ownership tracking.
> > >
> > >
> > > ## About Security Robustness
> > >
> > > In Section 3, preliminaries, we have defined the concept of security robustness. However, your suggestion is indeed valuable, and we should elaborate more on Security Robustness in the introduction section. We have made the relevant modifications to the paper, and the revised parts have been highlighted in red. Thank you very much for your suggestion.
> > >
> > > Furthermore, we extend our gratitude once again for your suggestions, which have significantly enhanced the quality of our paper. Additionally, would you consider revising your rating? Many thanks.

---

> > > > ### Comment · Reviewer_FTnC · 2023-11-22
> > > >
> > > > Thank you for your reply and for considering my suggestion. I appreciate your explanation of how ownership can be identified using your method. However, to make a convincing argument about ownership, a methodological explanation alone is not sufficient; supporting experiments are also necessary. If the experimental results do not provide support, the argument may be considered weak. Alternatively, narrowing the scope of the study could be another option. From the perspective of proactive method for generated language detection, your paper is well-written.
> > > >
> > > > If I have overlooked any experiments related to ownership in your study, could you please direct me to them?

---

> ### Author Response · Authors · 2023-11-22
> **Thank you for your patient and valuable feedback!**
>
> ## About Ownership Tracking
>
> There is indeed no experiment (or any content) about ownership tracking in our paper.  This is because the application of watermark algorithms to ownership tracking falls beyond the scope of this paper. Our objective is to present a semantically invariant, robust watermarking algorithm that achieves an optimal balance in terms of resistance to attacks and security robustness, compared to previous methods. We believe that elaborating on the integration of watermarking into ownership tracking systems is not necessary within this article.  Our previous description (in rebuttal) of how our method could perform ownership tracking was primarily to demonstrate the potential of our approach in this area.
>
> Thank you once again for your suggestion.
>
> ## update
>
> With less than three hours remaining until the end of the rebuttal period, if you have no further questions, would you consider updating your rating?

---

### Official Review · Reviewer_1hfF · 2023-11-01

**Soundness:** 3 good
**Presentation:** 3 good
**Contribution:** 3 good
**Rating:** 8
**Confidence:** 5

**Summary:**

This paper proposes a method for watermarking the outputs of autoregressive large language models. Following previous works, the proposed approach intervenes at the sampling step by perturbing the language model's predicted logits for the current token. Rather than determine the logit perturbation vector ("watermark logits") via a fixed partition of the token vocabulary or a hash of preceding tokens, the proposed method uses a "semantic" representation of preceding tokens to determine the perturbation. Because detection is performed analogously, the proposed method is robust against semantics-preserving transformations of watermarked text, while obtaining stronger security against the reverse-engineering attack of Sadasivan et al. [1].

[1] Vinu Sankar Sadasivan, Aounon Kumar, Sriram Balasubramanian, Wenxiao Wang, and Soheil Feizi. Can ai-generated text be reliably detected? arXiv preprint arXiv:2303.11156, 2023.

**Strengths:**

Using a "semantic" representation of previous tokens to determine the watermark logits for the current token, rather than e.g. a hash of the preceding n-gram, is a novel and interesting idea.

The general method of mimicking the green-list/red-list vocabulary partition of Kirchenbauer et al. [1] by using embedding and mapping networks to generate logit perturbations in $±1$ is clever, and the proposed training scheme for the mapping ("watermark") network is intuitive and well-motivated.

The proposed method achieves similar or better robustness to established language model watermarking methods while maintaining similar or better generated text quality (as measured by perplexity) and increased robustness against the reverse-engineering attack of Sadavisan et al.

[1] John Kirchenbauer, Jonas Geiping, Yuxin Wen, Jonathan Katz, Ian Miers, and Tom Goldstein. A watermark for large language models. arXiv preprint arXiv:2301.10226, 2023.

**Weaknesses:**

The detection equation (7) on pp.5 is not very clear. I'm assuming that detection is performed by computing the watermark logits vector for each time step, checking if the vector entry corresponding to the observed text token at each time step is 1 (as these logits should roughly be $±1$ given the saturated $tanh$), and keeping a running tally (in practice, a running sum of these selected logit values). This is basically how detection works with the watermark of Kirchenbauer et al., with the "green list" corresponding to watermark logits valued $1$ and the "red list" to logits valued $-1$. If this is indeed the case, it doesn't really come across in equation (7) -- the notation looks like we are taking the logit at the index corresponding to our current integer time step $j$ rather than the one corresponding to the observed token at the current time step $j$. I think either better notation or some additional explanation is needed.

While the authors consider the attack of Sadasivan et al., they do not consider _adaptive_ attacks performed with general knowledge of their proposed watermarking method. Could a motivated attacker leverage a semantic embedding model of their own to better estimate or remove the watermark? Even with access to only a surrogate embedder model and no access to the mapping/watermark model, my intuition is that the tight correspondence between semantic embeddings and watermark logits might make such an adaptive attack feasible. If the number of semantic contexts that need to be observed to infer the watermark's "rules" in such a scenario is much lower than, say, the number of n-grams that need to be observed for KGM-n, this would constitute a serious security flaw.

The evaluation of the quality of watermarked text is somewhat limited, only covering perplexity.

**Questions:**

If the semantic embeddings are insufficiently sensitive -- i.e., if they remain similar as more tokens are generated and added to the context -- couldn't this lead to highly repetitive generations as the same logits are boosted (similar semantic embeddings are mapped to similar watermark logits)? I could see this becoming an issue as the length of the context passed to the semantic embedder increases, as new tokens will constitute an increasingly small portion of the context window. The provided example of text generated with the proposed method (Appendix A, pp.12) seems more repetitive than the Kirchenbuaer et al. examples. It would be interesting to know whether the authors evaluated the quality of watermarked texts in terms of repetitiveness or other metrics beyond perplexity.

Have the authors tried using different fixed context lengths for the semantic embedder? Because the experiments in the paper use a generation length of only 200 tokens, the authors presumably feed all previous tokens to the semantic embedder. It would be interesting to see how short and long contexts (e.g. up to the limit of the embedding model) affect the performance of the watermark in terms of robustness, security, and text quality. The required context length could also have implications for the strength of the watermark as a function of the number of tokens generated, and its robustness to cut-and-paste attacks where generated text is spliced into human-written text [1].

The watermark of Kirchenbauer et al. allows for an arbitrary "green-list" size. With normalization, the distribution of watermark logits produced by the proposed method looks very symmetric (Figure 2c, pp.8), corresponding to a "green-list" size of 50% of the vocabulary. However, this is presumably computed in aggregate over the data distribution. In practice, do the watermark logits for individual time steps ever deviate significantly from an even split between $±1$?


[1] John Kirchenbauer, Jonas Geiping, Yuxin Wen, Manli Shu, Khalid Saifullah, Kezhi Kong, Kasun Fernando, Aniruddha Saha, Micah Goldblum, and Tom Goldstein. On the Reliability of Watermarks for Large Language Models. arXiv preprint arXiv:2306.04634, 2023.

---

> ### Author Response · Authors · 2023-11-19
>
> ## About the watermark detection method
>
> Please refer to the first response in "Response to all reviewers".
>
> ## About the potential adaptive attacks
>
> We understand your concern that users might be aware that our watermark insertion is related to the semantic embedding of the text. Therefore, we specifically demonstrate in Figure 3(a) the effect of a spoofing attack using text with highly concentrated semantic domain types(embeddings are more similar). While attacking the watermark with text of fine-grained topics like L2 class indeed shows some improvement in the attack's effectiveness, the enhancement is not significant. This, to some extent, indicates the sufficient security robustness of our method.
>
> Furthermore, for a user possessing an embedding model and attempting to break our watermark, designing such an algorithm is not straightforward and may constitute a highly complex task in algorithmic design. If you have a proposal, we are open to experimenting with it based on your approach.
>
> ## Text quality Evaluation
>
>
> To better validate the quality of the text generated by our watermarking method, we conducted further experiments in the field of machine translation. Specifically, we utilized the NLLB-200-distilled-600M model [Costa-jussà et al. (2022)] to perform experiments on the WMT14 dataset, focusing on two translation scenarios: French to English and German to English. The results of these experiments are as follows:
>
> | Setting | Mehod     | Ori.BLEU | Wat.BLEU | Detection F1 |
> |---------|-----------|----------|----------|--------------|
> | FR-EN   | KGW-1     | 37.9     | 36.5     | 99.8         |
> | FR-EN   | SIR(ours) | 37.9     | 36.8     | 1.0          |
> | DE-EN   | KGW-1     | 38.5     | 37.7     | 1.0          |
> | DE-EN   | SIR(ours) | 38.5     | 37.9     | 1.0          |
>
> Although there is a slight decrease in the BLEU score after watermarking, the extent of this decrease is minimal. This indicates that the impact of our method on the quality of the text is smaller compared to that of the KGW-1 approach. We have updated the result in appendix G.
>
> ## About the repetitiveness
>
> With increasing text length, the degree of variation in embeddings diminishes. In the most extreme scenario, if the addition of any token does not alter the embedding, our method essentially degenerates to KGW-1 (a global red-green list). Consequently, the texts produced by our approach theoretically exhibit less repetition compared to KGW-1. To examine the repetition issue, we conducted statistical experiments on the N-gram repetition ratio in generated texts. The results are as follows:
>
>
> | Method    | 1-gram | 2-gram | 3-gram |
> |-----------|--------|--------|--------|
> | SIR(ours) | 0.41   | 0.11   | 0.02   |
> | KGW-1     | 0.46   | 0.14   | 0.03   |
> | KGW-2     | 0.40   | 0.09   | 0.02   |
> | KGW-4     | 0.38   | 0.07   | 0.01   |
>
>
> The level of repetition in our generated texts is lower than that of KGW-1, but higher than KGW-2 and KGW-4. Although our method did not achieve the lowest degree of repetition, compared to the KGW series, it still represents an optimal balance in terms of text repetitiveness, robustness, and security. We have updated the result in appendix I.
>
> ## About the context length
>
> A longer context length can enhance the effectiveness, yet a shorter context length results in a decrease in the overall watermark detection performance and robustness. However, this is an inherent limitation of watermark algorithms not only our methods. Here we present a comparison of the effects at different lengths.
>
> | Method    | 50-L  Ori/Re | 100-L Ori/Re | 300-L Ori/Re | 600-L Ori/Re |
> |-----------|--------------|--------------|--------------|--------------|
> | SIR(ours) | 92.8/78.4    | 98.4/87.5    | 100/92.2     | 100/98.7     |
> | KGW-1     | 92.5/77.3    | 97.5/88.0    | 100/92.4     | 100/98.9     |
> | KGW-2     | 91.7/71.5    | 98.0/81.2    | 100/84.8     | 100/93.2     |
> | KGW-4     | 92.3/65.2    | 98.1/71.4    | 100/73.5     | 100/82.1     |
>
> The shows that as the generation length increases, both the effectiveness and robustness of detection improve. This trend is consistent with the KGW. Even when the length exceeds 600, surpassing the 512-length limit of the embedding model, truncating the context does not affect the specific detection. We have updated the result in appendix H.
>
> ## About the copy paste attack
> Please refer to the third response in "Response to all reviewers".
>
> ## About the variance of watermark logits distribution
>
> Your concern is valid, but in reality, our statistical analysis has found that in all steps, the watermark logits are consistently divided into two parts, 1 and -1, without any significant deviation. We have calculated the proportion of 1 and -1, finding the mean to be 0.5 with a standard deviation not exceeding 0.01.
>
> ## Reference
>
> Kirchenbauer,et al."A watermark for large language models."
>
> Costa-jussà, et al."No language left behind: Scaling human-centered machine translation."

---

> ### Comment · Reviewer_1hfF · 2023-11-22
> **Response to authors**
>
> I thank the authors for their reply and for performing additional experiments. I think the inclusion of these experiments, particularly the additional attacks, will strengthen the paper. I'm willing to raise my score, but I have a three points that I'd like addressed/clarified:
>
> __1.__ In the final draft it would be good to see the additional experimental results cleaned up a bit, with the new attacks moved into Table 1 if at all possible. I also think some additional details on the copy-paste attack are necessary -- e.g. how "topics" are defined.
>
> __2.__ I think the paper could make the connection between KGW and the proposed method even more explicit. For example, the authors could highlight how the {±1} logit perturbations essentially define red/green lists, and how detection is essentially performed in the same manner as KGW using these "relaxed" red/green lists.
>
> __3.__ The paper's central idea -- extending the "red-list/green-list" approach of KGW to enable more general conditional control over logit perturbations via embedding and mapping networks -- is novel and interesting. It allows for replacing the n-gram hashing of KGW with any suitable text embedding, and appears to offer some flexibility over "red-list/green-list" distributions through the training objectives of the mapping network. It would be nice if the authors could remark on this potential generality -- i.e., whether the authors expect that the proposed method only works within a narrow range of configurations (CBERT/SBERT sentence embeddings mapped to ~50/50 ±1 logit distributions), or whether it might be worth exploring other configurations in future work. The embedder experiments in section C suggest some rough guidelines (e.g. "near-normal" distribution of embedding distances), but I think the generality/flexibility angle is interesting enough to merit addressing directly, even if only briefly.

---

> > ### Author Response · Authors · 2023-11-22
> > **Thanks for your valuable feedback**
> >
> > ## About the additional experimental results and definition of topic
> >
> > Thank you for your suggestion. We have moved the results of the copy-paste attack to Table 1. Since the base LLM (llama2-chat) used for the emoji attack differs from the other data in Table 1 (llama2), we keep it it in appendix G.
> >
> > Regarding the concept of 'topic' having the same topic implies that the LLM continues a text with 150 tokens corresponding to the original 600 tokens. Having different topics means that after the LLM generates 150 tokens, the original 600 tokens are replaced with other 600 tokens from different text . We have provided additional explanations in Appendix G.
> >
> > ## Make more clear connection
> >
> > Thank you very much for your suggestion. We have provided clearer descriptions of the relevant concepts in sections 4.2, 4.3, and Appendix D.
> >
> > ## Other configuration of watermark model
> >
> > In Appendix D, we have supplemented information on setting different ratios of 1 and -1 (other than 50%) for watermark logits generated by our watermark model. We demonstrate its feasibility and present images of watermark logits outputs for various ratios.
> >
> > In conclusion, we are immensely grateful for your valuable suggestions, which have significantly contributed to enhancing the quality of our paper.

---

> > > ### Comment · Reviewer_1hfF · 2023-11-22
> > > **Response to authors**
> > >
> > > I thank the authors for addressing my comments. I have adjusted my score correspondingly.
> > >
> > > A few notes for the final draft:
> > >
> > > * The copy-paste attack should be updated to match at least a couple of the configurations used by Kirchenbauer et al. [1] (Section 4.2), i.e. using a fixed number of insertions and proportion of watermarked text rather than simply prepending a fixed number of tokens. Kirchenbauer et al. do this at passage lengths as small as 200 tokens, so it should be feasible.
> > >
> > > * The authors should either re-do the emoji attack with the same language model used in other attacks and incorporate the results into Table 1, or explain why a different model was used.
> > >
> > > [1] John Kirchenbauer, Jonas Geiping, Yuxin Wen, Manli Shu, Khalid Saifullah, Kezhi Kong, Kasun Fernando, Aniruddha Saha, Micah Goldblum, and Tom Goldstein. On the reliability of watermarks for large language models. arXiv preprint arXiv:2306.04634, 2023.

---

> > > > ### Author Response · Authors · 2023-11-23
> > > > **Thanks again for your patient and valuable feedback!**
> > > >
> > > > Thank you for all the suggestions provided. We will update our final draft to include more experiments on copy-paste attacks in more detailed configurations, and also add additional experiments to Table 1.
> > > >
> > > > The sole reason Emoji attacks were not included in Table 1 now is that the data there was derived from experiments conducted with the non-instruct-tuned LLaMA2 model. Emoji attacks are stable only on instruct-tuned models, such as LLaMA2-chat, which can comprehend complex instructions. We will re-conduct all experiments in Table 1 with  LLaMA2-chat and update the final draft with a unified experimental table.
> > > >
> > > > Finally, we would like to express our sincere gratitude for your efforts during the review process, which have been immensely beneficial to us.

---

### Author Response · Authors · 2023-11-19
**Response to all reviewers**

## About the detection method

Firstly, there are indeed some confusing parts in our detection algorithm, and we appreciate reviewer 1hfF for pointing out that typo $ P_{\mathrm{W}}^{(j)} $ should be expressed as $ P_{\mathrm{W}}^{(t_j)} $, denoting the logit at the selection of token $ t_j $ at the $j$th step.

Regarding the concern raised by yZti that our detection not providing a stringent statistical affirmation concerning the rate of false positives, we must highlight that our detection algorithm fundamentally aligns with the approach of KGW (2023). This is because the value of each token in the watermark logits calculated from $ P_{\mathrm{W}}^{(t_j)} $ is close to either 1 or -1, corresponding to the green and red tokens in KGW (2023). Both our method and KGW (2023) determine the presence of a watermark by counting the number of watermark logits close to 1 (the number of green tokens).

However, the expression in our original paper is indeed not very clear. Both our detection score and the method by KGW (2023) involve conducting a z-value test. Specifically, we perform this test on the average logits value for each token. Given that the mean of this value is zero and its variance is one, the formula for the z-value test is identical to that for calculating the mean value:

$$
z =  \frac{\sum_{j=1}^{N} (P_{\mathrm{W}}^{(t_j)}(x^{prompt}, \boldsymbol{t}_{:j-1}) - 0)}{N*1}
$$

$$
= \frac{\sum_{j=1}^{N} P_{\mathrm{W}}^{(t_j)}(x^{prompt}, \boldsymbol{t}_{:j-1})}{N}.
$$

We have changed the expression to make it more clear in the updated version.

About the resource consumption of the detection process. Indeed, we use an embedding model for detection compared to KGW (2023), but our computation process is not slow. Since the embedding of a sentence does not undergo drastic changes with the addition of a token, we can calculate the embedding at intervals of a trunk size. Additionally, the computation of embeddings can be parallelized. In our tests, the detection of a 200-token text can be completed within 2 seconds on a 3090 GPU. Furthermore, it's not always necessary to make watermark detection available to users, which could expose the watermark generation method and lead to watermark forgery.

## About the comparision of our method and KGW-1

Reviewers 1s48 and yZti noted that our performance under the beam search setting in Table 1 is slightly inferior to KGW-1. We acknowledge that KGW-1 indeed demonstrates robustness in adversarial text modification. However, our primary contribution lies in achieving comparable adversarial robustness to KGW-1 while obtaining much higher security robustness. For details, refer to the explanation in Figure 3(a).

## Robustness to more attack type

We acknowledge that even though our work has implemented numerous attack methods, including text rewriting, synonym replacement, and spoofing attacks, to test the robustness, there are still some missing attacks.  We here conduct two additional attack methods mentioned by the reviewers: the emoji attack and the copy-paste attack (prefix injection attack).  We have updated the result in appendix G.

Regarding the emoji attack as described by Kirchenbauer et al., we conducted experiments using the llama2-7b-chat model. Here are the specific experimental results:

| Method    | Origin F1 | Emoji attack F1 |
|-----------|-----------|-----------------|
| SIR(ours) | 100       | 98.6            |
| KGW-1     | 99.7      | 99.0            |
| KGW-2     | 100       | 53.2            |
| KGW-4     | 100        | 49.7            |

Both our method and KGW-1 exhibit strong robustness against the emoji attack. The robustness of KGW-1 is attributed to its generation of a global red-green list, which minimizes the impact of inserted emojis. On the other hand, our SIR method demonstrates significant resilience because the insertion of emojis does not drastically alter the sentence's embedding. Of course, this depends on the robustness of the embedding model, but in our experiments, our method proved to be highly robust to the emoji attack.

We further explored the robustness against the copy-paste attack. Following the approach of Kirchenbauer et al., we inserted 150 watermarked tokens into 600 human tokens. Specifically, we tested two scenarios: the first where the human text and watermarked text share the same context, meaning they convey the same topic; the second scenario involved entirely different contexts, where the human text and watermarked text address distinct topics. The experimental results are as follows:
| Method    | Same topic |  Different Topic |
|-----------|---------------|--------------------|
| SIR(ours) | 91.8          | 88.1               |
| KGW-1     | 92.0          | 91.8               |

In scenarios where the topics are the same, the robustness of both our method and the KGW-1 method is nearly identical. However, in cases involving different topics, the effectiveness of our method may slightly diminish.

---

### Meta-Review · Area_Chair_zZK2 · 2023-12-05

**Metareview:**

Previous strategies for the task of watermarking of modern generative language model outputs describe watermarks that are non-semantic, and in almost all cases directly seeded by text n-grams. This opens up several avenues for attack and fundamentally limits the robustness of these watermarks in the face of non-semantic changes of the watermarked text. Reducing the n-gram size is also not an option, as this directly increases the detectability of the watermark.

This submission instead describes a general strategy toward encoding semantic watermarks, which can circumvent these problems with n-gram based non-semantic watermarks. The submission then trains a small watermarking model based on several considerations that is used to encode the watermark in model outputs. This approach is evaluated against several strategies that attack or modify generated text and compared to a number of previous approaches to text watermarking.

Overall I consider this submission to be of definite interest to the community. The initial results in this work may only represent the first step toward truly semantic watermarks, but certainly an interesting one.


On a final note, a remaining weakness during the reviewer discussion that was brought up was whether the proposed approach could guarantee certain false positive rates analytically. From my reading of the paper and the authors' comments, this is not the case. The watermarking model is optimized so that the null hypothesis is approximately true on non-watermarked text, but no guarantee can be provided (e.g. on out-of-domain text). I do not consider this to be a critical problem, but I require the authors to amend section 4.3 in the camera-ready version to clarify this for interested readers and follow-up work.

**Justification For Why Not Higher Score:**

The proposed watermark is sound, yet the actual gain in robustness as showcased in the experiments in this submission is lower than anticipated, limiting the immediate impact of this work.

**Justification For Why Not Lower Score:**

The approach is innovative, and presents an innovative generalization of existing n-gram based watermarks toward generic encoders that is likely to find use and further development in the future.

---

### Decision · Program_Chairs · 2024-01-16

Accept (poster)